# Hybrid Lattice-Boltzmann-Potential Flow Simulations of Turbulent Flow around Submerged Structures

**Christopher M. O'Reilly** [1,2,*], **Stephan T. Grilli** [1,*], **Christian F. Janßen** [3], **Jason M. Dahl** [1] and **Jeffrey C. Harris** [4]

[1] Department of Ocean Engineering, University of Rhode Island, Narragansett, RI 02882, USA
[2] PacMar Technologies, South Kingstown, RI 02879, USA
[3] Institute for Fluid Dynamics and Ship Theory, Hamburg University of Technology, 21071 Hamburg, Germany
[4] LHSV, Ecole des Ponts, EDF R&D, 78400 Chatou, France
[*] Correspondence: coreilly@pacmartech.com (C.M.O.); grilli@uri.edu (S.T.G.)

**Abstract:** We report on the development and validation of a 3D hybrid Lattice Boltzmann Model (LBM), with Large Eddy Simulation (LES), to simulate the interactions of incompressible turbulent flows with ocean structures. The LBM is based on a perturbation method, in which the velocity and pressure are expressed as the sum of an inviscid flow and a viscous perturbation. The far- to near-field flow is assumed to be inviscid and represented by potential flow theory, which can be efficiently modeled with a Boundary Element Method (BEM). The near-field perturbation flow around structures is modeled by the Navier–Stokes (NS) equations, based on a Lattice Boltzmann Method (LBM) with a Large Eddy Simulation (LES) of the turbulence. In the paper, we present the hybrid model formulation, in which a modified LBM collision operator is introduced to simulate the viscous perturbation flow, resulting in a novel *perturbation* LBM (pLBM) approach. The pLBM is then extended for the simulation of turbulence using the LES and a wall model to represent the viscous/turbulent sub-layer near solid boundaries. The hybrid model is first validated by simulating turbulent flows over a flat plate, for moderate to large Reynolds number values, Re $\in [3.7 \times 10^4; 1.2 \times 10^6]$; the plate friction coefficient and near-field turbulence properties computed with the model are found to agree well with both experiments and direct NS simulations. We then simulate the flow past a NACA-0012 foil using a regular LBM-LES and the new hybrid pLBM-LES models with the wall model, for Re = $1.44 \times 10^6$. A good agreement is found for the computed lift and drag forces, and pressure distribution on the foil, with experiments and results of other numerical methods. Results obtained with the pLBM model are either nearly identical or slightly improved, relative to those of the standard LBM, but are obtained in a significantly smaller computational domain and hence at a much reduced computational cost, thus demonstrating the benefits of the new hybrid approach.

**Keywords:** hybrid method; lattice-boltzmann method; potential flow; GPGPU implementation

## 1. Introduction

Numerical models simulating the irrotational motion of an incompressible, inviscid fluid, based on potential flow theory, are computationally efficient and often sufficiently accurate to simulate many large scale fluid–structure interaction problems in ocean engineering, including those involving free surface waves (e.g., [1]). However, potential flow models cannot be used in applications where viscous effects are important– for instance, in the boundary layer near solid boundaries, in the wake of bluff bodies, or when considering surface wave breaking. Standard Computational Fluid Dynamics (CFD) Navier–Stokes (NS) solvers, such as based on a finite volume, e.g., [2–4] or Lattice Boltzmann (LBM) e.g., [5–10] method, can accurately model these types of flows, but are typically computationally costly. Additionally, for free surface flows, NS solvers are often too numerically dissipative to model wave propagation over long distances e.g., [3].

An alternative hybrid modeling approach, to accurately and more efficiently solve fluid–structure interaction problems of interest to ocean and other engineering disciplines,

has been proposed in earlier work based on a *Helmholtz decomposition* method [11,12]. Unlike one- or two-way coupled models that have separate regions of the computational domain matched along a common boundary e.g., [3,13]; in this method, both the velocity and pressure fields are expressed as the sum of inviscid/irrotational (*I*) and viscous perturbation (*P*) components. Each component of the decomposition is solved using different numerical models and methods in separate, but *overlapping* , computational domains. In this decomposition approach, the *P* fields are governed by a modified (perturbed) NS equation, forced by the *I* fields, obtained from the inviscid solution. This approach was previously successfully used by Harris and Grilli [4] to model turbulent flows over solid boundaries using a finite volume method, and validated for turbulent channel and wave induced boundary layer flows. It was also used to simulate viscous effects in linear ship seakeeping by Reliquet et al. [14].

Following this approach, we develop and apply a new hybrid model in which the *I* fields, governed by potential flow equations, are solved (analytically or numerically) over the complete, larger size, computational domain that includes the structure of interest and extends to the *far-field* , and the *P* fields are solved with a LBM model in a smaller *near-field* domain around the structure. The latter domain covers the region of the flow in which viscous/turbulent effects are deemed to be important based on the considered problem (this will be made more clear in applications). Hence, the more computationally demanding LBM model is only applied to the *P* fields in the smaller near-field domain, yielding a more computationally efficient solution than applying a LBM model to the entire domain, while ensuring that the complete NS solution is solved where the physics calls for it. The coupling between continuum mechanics-based Equations (or models), such as potential flow, and the kinetic-based LBM is less straightforward than earlier implementations of the hybrid method based on a volume of fluid NS solver [4]. In particular, as will be shown in the paper, one must derive a perturbation LBM equivalent to the nonlinear $I - P$ coupling terms that appear in the perturbation NS equations.

The presented perturbation method has computational benefits from using the LBM to solve the NS equations. Compared to more standard finite volume solvers, the data locality and kernel simplicity of the LBM, based on a weakly compressible approach, allow for a very efficient parallel implementation of the model, particularly on "General Purpose Graphical Processor Units" (GPGPU) [15–17], while a single GPGPU still has a limited memory (although new and increasingly powerful hardware is regularly designed; e.g., the 2020 A100 NVDIA GPU has 40 or 80 GB memory, nearly 7000 cores and a 20 teraflop performance) a multi-GPGPU implementation of the LBM may allow achieving a higher computational efficiency, for an identical accuracy, than traditional CFD solvers implemented on a massively parallel CPU cluster. In the hybrid method context, for many engineering applications, the smaller computational domain size where the pLBM is solved can often be simulated using a single GPGPU e.g., [18], allowing simulations to be run on a desktop computer equipped with a relatively inexpensive GPGPU co-processor. When the potential flow is also solved with a numerical model, e.g., BEM based, its solution may then be calculated using the computer's often parallelized CPUs, with limited conflicting resource requirements. If a traditional NS solver were to be used in place of the LBM, a much larger number of CPU cores would be required to run it at an accuracy equivalent to that of the LBM.

In engineering applications involving complex boundary conditions and/or boundary/structure geometry, the potential flow solution over the entire computational domain must also be efficient. To this effect, a generic numerical solver, such as that based on the higher-order Boundary Element Method (BEM), have been used that feature fully nonlinear free surface boundary conditions if applicable e.g., [4,19]. For simulating fully nonlinear wave-structure interactions in large three-dimensional (3D) domains, efficient BEM solvers with a parallelized Fast Multipole Algorithm (FMA) have been developed [20]. Cases with a free surface are not considered in the present paper, but have been reported on elsewhere. To assess the ability of the LBM to simulate strongly nonlinear free surface flows,

Janssen et al. [9,15,19] simulated 2D wave breaking problems already simulated with other methods [3,13], using a weak coupling approach between potential flow and a LBM in which a Volume Of Fluid (VOF) interface tracking method was used. In such cases, the LBM model was simply initialized with potential flow results for waves that had been propagated up to close to the breaking point in a potential flow BEM model [21–23]. Next, Janssen et al. computed similar results with the hybrid method, in which the $I - P$ coupling terms were represented as LBM body force terms, using the pre-computed $I$ fields to force the $P$ field solution through these terms. This approach, while proven effective, required computing spatial derivatives of both the $I$ and $P$ fields using finite difference approximations that yielded a compact but non-local LBM kernel, as well as higher truncation errors than in the original LBM collision operator. Both of these reduced the overall efficiency and accuracy of the parallelized GPGPU solution. Therefore, Janssen [9] suggested that instead, the nonlinear $I - P$ coupling terms could directly be introduced into the LBM equilibrium probability distribution functions (EPDFs), hence, to develop *perturbation* EPDFs or pEPDFs. The latter were incrementally developed, implemented, and validated as part of the development of a perturbation LBM (pLBM) model component to a hybrid naval hydrodynamic solver by [18,24–26], in which the potential flow solution, with fully nonlinear free surface boundary conditions (FNPF), was computed using a higher-order BEM-FMA model [20].

As an additional contribution to this line of work, the present paper details the development of a hybrid potential flow-LBM model for external turbulent flows around submerged structures. This development extends the pLBM approach proposed earlier by O'Reilly et al. [26] to high-Reynolds number flows, using large eddy simulations (LES) and a turbulent wall model. This new model is validated on a few applications for which the potential flow fields $I$ can be solved analytically to force the pLBM solution. Cases with a more complex geometry could be easily solved later with the same model, using instead a numerical solution of the $I$ field with the BEM-FMA model mentioned earlier.

In the paper, we first describe the pLBM formulation with a Multiple Relaxation Time (MRT) collision operator and introduce the sub-grid turbulence scheme by modifying the LBM-LES model proposed by Krafczyk et al. [10] to apply to the pLBM. To improve the representation of turbulent boundary layers near solid boundaries without the need for a refined discretization, a wall model approach is added to the simulations. Recent efforts on boundary layer flow simulations with the more advanced Cumulant LBM are presented for a wall model simulation of atmospheric boundary layers [27], for flat plat boundary layer transitions [28], and for periodic hill flow [29]. Here, a pLBM wall model that is based on the work of Malaspinas and Sagaut [30] is presented. Some modifications were made to the wall model to improve its accuracy for highly curved boundaries of arbitrary shape and orientation. The LBM-LES with the wall model and its pLBM counterpart were validated in terms of convergence and accuracy by simulating turbulent channel flows for which there are reference solutions e.g., [30]. The method was then applied and validated for a more demanding test, by computing the drag and lift forces, and pressure distribution on a NACA0012 foil at a few Reynolds numbers, up to a large value $Re = 1.44 \times 10^6$, using both the standard LBM and perturbation LBM. Results were again compared to reference data, and with each other, leading to a discussion of both methods' performance.

## 2. The Lattice Boltzmann Method

In the LBM, the weakly compressible NS Equations (assuming a low Mach number **Ma**) are modeled by solving an equivalent mesoscopic problem, in which the fluid is represented by particles interacting over a (typically regular) lattice (or grid). The main LBM variables are the particle distribution functions (DFs) $f_\alpha(t, \mathbf{x}, \mathbf{e}_\alpha)$, which represent the normalized probability of finding particle $\alpha$ at location $\mathbf{x}$ and time $t$, moving with velocity $\mathbf{e}_\alpha$. Once the DFs are computed at time $t$, the macroscopic hydrodynamic quantities, i.e., velocity $\mathbf{u}$ (or $u_i$) and density $\rho$ are found as moments of the DFs, and the pressure is expressed as $p = p_0 + (\rho - \rho_0)c_s^2$, where $p_0$ and $\rho_0$ are the ambient pressure and associated

density (set to zero in the following), while $c_s$ denotes the speed of sound in the considered medium (see later for details).

### 2.1. Summary of Macroscopic Flow Equations

The macroscopic equations solved in the LBM are the standard mass and momentum conservation NS Equations (using the summation convention of tensor notation; $i, j = 1, 2, 3$),

$$\frac{\partial u_i}{\partial x_i} = \mathcal{O}(\Delta x^2, \Delta t^2) + \mathcal{O}(\mathbf{Ma}^2) \tag{1}$$

$$\rho \left\{ \frac{\partial u_i}{\partial t} + u_j \frac{\partial u_i}{\partial x_j} \right\} = \frac{\partial \tau_{ij}}{\partial x_j} + \rho \, G_i + \mathcal{O}(\Delta x^2, \Delta t^2) + \mathcal{O}(\mathbf{Ma}^2), \tag{2}$$

where $\tau_{ij}$ is the stress tensor and $G_i$ the body force acceleration (usually gravity). For an isothermal fluid, a Chapman-Enskog expansion (see Appendix A) would show that LBM equations approximate NS equations up to second-order in grid spacing $\Delta x$, time step $\Delta t$, and Mach number $\mathbf{Ma} = U/c_s$, with $U$ a characteristic flow velocity e.g., [31]. For incompressible Newtonian fluids, we further have $\tau_{ij} = -p \, \delta_{ij} + 2\mu S_{ij}$, with $\mu$ the dynamic viscosity ($\nu = \mu/\rho$, the kinematic viscosity) and $S_{ij}$ the rate of strain tensor defined later. For small Mach numbers $\mathbf{Ma} \ll 1$, $(\rho - \rho_0)/\rho_0 \ll 1$, with $\rho_0$ a reference (average) density; hence, the fluid can be considered as nearly incompressible [32].

When modeling turbulent flows at high Reynolds numbers, the velocity and pressure are usually decomposed into mean and fluctuation components, i.e., $u_i = \overline{u_i} + u_i'$ and $p = \overline{p} + p'$ (with overbars indicating time averaging). Introducing this decomposition into Equations (1) and (2) and averaging them formally yields identical equations for the mean flow variables, referred to as Reynolds averaged NS Equations (RANS), with an additional Reynolds stress term, $R_{ij} = -\rho \overline{u_i' u_j'}$, in the stress tensor, representing effects of turbulent momentum exchanges. Note, in the context of LES modeling, which is introduced later, it is more accurate to refer to the averaged Equations (1) and (2) as filtered NS equations and to $R_{ij}$ as subgrid scale (SGS) stresses.

In the standard LES model, the deviatoric part of the Reynolds stresses, $R_{ij}^d = R_{ij} - \frac{1}{3} R_{kk} \delta_{ij}$ (note, $\frac{1}{3} R_{kk} = \frac{2}{3} \rho k$, where $k$ here denotes the turbulent kinetic energy), is parameterized by a SGS turbulent closure model as $R_{ij}^d = \mu_T \, \overline{S}_{ij}$, with $\mu_T$ the eddy viscosity, a function of flow parameters, and $\overline{S}_{ij}$ the mean (resolved) rate of strain tensor (see details later). For gravity forces we further have, $\rho \, G_i = -\frac{\partial}{\partial x_i}(\rho g x_3)$ (with $g$ the gravitational acceleration) and, defining the resolved turbulent dynamic pressure as, $\tilde{p} = \overline{p} + \rho g x_3 - \frac{2}{3} \rho k$, the resolved turbulent shear stress tensor to use in the right-hand-side (RHS) of filtered Equation (2) without gravity forces reads, $\tilde{\tau}_{ij} = \tilde{p} \, \delta_{ij} + (\mu + \mu_T) \, \overline{S}_{ij}$.

Overbars will be dropped in the following as filtered LES equations will be those used to represent mean (resolved) turbulent flows.

### 2.2. Lbm Basics

The time evolution of the discrete particle DFs is governed by the Boltzmann advection-collision equation,

$$\frac{Df_\alpha}{Dt} = \frac{\partial f_\alpha(t, \mathbf{x})}{\partial t} + \mathbf{e}_\alpha \cdot \frac{\partial f_\alpha(t, \mathbf{x})}{\partial \mathbf{x}} = \Omega_\alpha + B_\alpha, \tag{3}$$

in which, $\mathbf{e}_\alpha$ denotes the discrete velocity vector of particle $\alpha$, $\Omega_\alpha$ is a collision operator describing interactions between particles, and $B_\alpha$ represents effects of volume forces. Equation (3) is discretized over a regular 3D lattice of grid spacing $\Delta x = \Delta y = \Delta z$, here following the standard D3Q19 scheme (Figure 1), using $n = 19$ discrete particle velocities and respective directions referred to as $\alpha = 0, \ldots, 18$, with $\alpha = 0$ denoting the reference particle lattice location. The corresponding velocity vectors: $\mathbf{e}_\alpha = \{0, 0, 0\}; \{\pm c, 0, 0\}; \{0, \pm c, 0\}; \{0, 0, \pm c\}; \{\pm c, \pm c, 0\}; \{\pm c, 0, \pm c\}; \{0, \pm c, \pm c\}$, for $\alpha = 0, \ldots, 18$, point in the directions of 18 neigh-

boring particles from the reference particle location. The particle propagation speed on the lattice is defined as $c = \Delta x / \Delta t$, which means that each particle travels the length of one lattice grid cell over one time step. Note, when a LBM model with nested meshes is used, the lattice resolution is refined within some sub-regions of the flow, such as near structures e.g., [26].

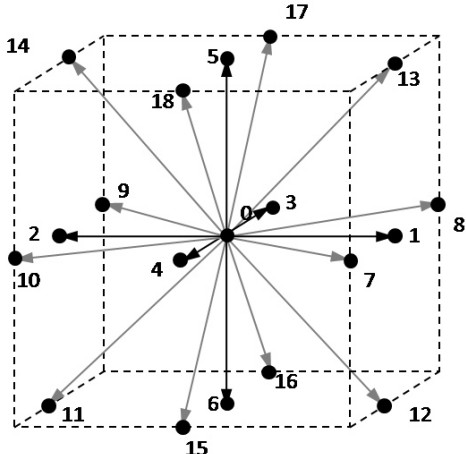

**Figure 1.** D3Q19 sub-lattice LBM, where vectors indicate $\alpha = 1, \ldots, 18$ directions (and length) for a particle to move on the lattice.

In the standard single relaxation time (SRT) LBM, Equation (3) is discretized by finite differences in space and time yielding,

$$f_\alpha(t + \Delta t, \mathbf{x} + \mathbf{e}_\alpha \Delta t) = f_\alpha(t, \mathbf{x}) - \frac{\Delta t}{t_0} \{ f_\alpha(\mathbf{x}, t) - f_\alpha^{eq}(\rho, \mathbf{u}) \} + \Delta t \, B_\alpha, \tag{4}$$

where $f_\alpha^{eq}(\rho, \mathbf{u})$ are the equilibrium DFs (towards which the DFs converge), defined later as a function of $\rho$ and $\mathbf{u}$, and $\Omega_\alpha = (f_\alpha^{eq} - f_\alpha)/t_0$, with $t_0$ the single relaxation time. In the SRT method, the effect of a homogeneous body force such as gravity is typically expressed as, $B_\alpha = w_\alpha \mathbf{e}_\alpha \cdot \mathbf{G}_\alpha / c_s^2$ [33], where $\mathbf{G}_\alpha$ denotes the gravity force applied to particle $\alpha$. LBM simulations are split into a nonlinear *collision* step, which locally drives the particle DFs to their equilibrium values, and a linear *propagation* step, during which the updated DFs are advected [7].

For the LBM solution to converge towards that of the incompressible (i.e., weakly compressible with the use of acoustic scaling between the discretized spatial and temporal step size) NS Equations (1) and (2), the equilibrium function (5) must be defined as in He and Luo [7].

$$f_\alpha^{eq}(\rho, \mathbf{u}) = w_\alpha \left( \rho + \rho_o \left( 3 \frac{(\mathbf{u} \cdot \mathbf{e}_\alpha)}{c^2} + \frac{9}{2} \frac{(\mathbf{u} \cdot \mathbf{e}_\alpha)^2}{c^4} - \frac{3}{2} \frac{\mathbf{u}^2}{c^2} \right) \right), \tag{5}$$

To achieve the stated truncation errors, the expansion in (4) must be defined with $t_0 = 3\nu/c^2 + \Delta t/2$ and $\rho_o$, the average fluid density, where $\rho - \rho_0$ is a small perturbation resulting from the weak fluid compressibility. In the D3Q19 lattice scheme, isotropy is maintained by applying direction dependent weights $w_\alpha$ to the equilibrium DFs, i.e., $w_0 = 1/3$, $w_{1\ldots6} = 1/18$ and $w_{7\ldots18} = 1/36$, and defining the speed of sound in the medium as $c_s = c/\sqrt{3}$. Once the LBM equations are solved for the particle DFs, the macroscopic hydrodynamic quantities are found at each lattice node as moments of the DFs as,

$$\rho = \sum_{\alpha=0}^{n-1} f_\alpha, \qquad \rho_o u_i = \sum_{\alpha=0}^{n-1} e_{\alpha i} f_\alpha. \tag{6}$$

When modeling high Reynolds number flows, however, earlier work has shown e.g., [5] that more accurate and stable results can be obtained using the multiple relaxation time (MRT) LBM. In this method, the collision operator is defined as a function of higher-order moments of macroscopic flow properties that each have different physical meaning [34]. Using the D3Q19 lattice scheme, 19 equilibrium moments $m_\alpha$ are thus defined, some of which will be used to implement the LES in the LBM, allowing one to simulate high Reynolds number turbulent flows. The MRT collision operator is defined as ($\alpha, \beta, \gamma = 0, \ldots, 18$),

$$\Omega_\alpha = -M_{\alpha\gamma}^{-1} D_{\gamma\delta} (M_{\beta\delta} f_\beta - m_\delta^{eq}), \tag{7}$$

and is replaced in Equation (4), where $M_{\alpha\gamma}$ denotes the transformation matrix from DFs to moments $m_\gamma$, with $f_\alpha = M_{\alpha\gamma}^{-1} m_\gamma$ and $D_{\gamma\delta}$ is a diagonal collision matrix of relaxation parameters that allows applying different weights to different moments/fluid properties e.g., [31]. The equilibrium moments corresponding to Equation (5) and the NS equations are found as,

$$m_0^{eq} = \rho, \quad m_3^{eq} = \rho u_x, \quad m_5^{eq} = \rho u_y, \quad m_7^{eq} = \rho u_z$$
$$m_1^{eq} = e^{eq} = \rho_0(u_x^2 + u_y^2 + u_z^2), \quad m_9^{eq} = 3p_{xx}^{eq} = \rho_0(2u_x^2 - u_y^2 - u_z^2)$$
$$m_{11}^{eq} = p_{zz}^{eq} = \rho_0(u_y^2 - u_z^2), \quad m_{13}^{eq} = p_{xy}^{eq} = \rho_0(u_x u_y)$$
$$m_{14}^{eq} = p_{yz}^{eq} = \rho_0(u_y u_z), \quad m_{15}^{eq} = p_{xz}^{eq} = \rho_0(u_x u_z), \tag{8}$$

where non-listed moments are set to zero, i.e., $m_2$ representing the kinetic energy, $m_{4,6,8}$ related to the heat fluxes, $m_{10,12}$ representing the fourth-order moments, and $m_{16,17,18}$ the third-order moments. Additional details of the transformation matrix and the exact choice of relaxation rates in the collision operator may be found in [9].

In the LBM, it is customary to scale the physical variables using spatial, temporal, and mass scales: $\lambda$, $\Theta$, and $\varpi$, respectively, and define non-dimensional lattice variables (denoted by primes in the following). Thus, for the mesh parameters we have, $\Delta x' = \Delta x / \lambda$, $\Delta t' = \Delta t / \Theta$, $c' = c\Theta / \lambda$, and $m' = m / \varpi$. It is also customary to assume that $c' = 1$, which is equivalent to having a unit mesh Courant number. If the length scale is further defined as $\lambda = \Delta x$, we have $\Delta x' = 1$, which also requires that $\Theta = \Delta t$ and, hence, $\Delta t' = 1$. With these definitions, $c_s' = 1/\sqrt{3}$ and, in the SRT, $t_0' = 3\nu' + 1/2$, with $\nu' = \nu\Theta/\lambda^2$. In applications, scaled flow properties will be specified by their Mach number $\text{Ma} = U/c_s = U'/c_s'$ and Reynolds number, $\text{Re} = U\ell/\nu = U'\ell'/\nu'$ (with $\ell$ a representative length scale of the flow). Combining the various definitions we find, $U' = \text{Ma}/\sqrt{3}$ and $\text{Ma} = \sqrt{3}\,\text{Re}\,\nu'/\ell'$. For simplicity, in the following, we will drop the prime notation and use non-dimensional lattice variables in the LBM, unless stated otherwise.

*2.3. Macroscopic Equations for the Perturbation LBM*

We first recap the equations of the NS perturbation method used in earlier work e.g., [4,12,26] and develop the corresponding LBM equations, referred to as perturbation LBM (pLBM), with a MRT approach. Applying a Helmholtz decomposition to the flow, both the resolved velocity $u_i$ and dynamic pressure $\tilde{p}$ are expressed as,

$$u_i = u_i^I + u_i^P \qquad \text{with} \qquad \tilde{p} = \tilde{p}^I + \tilde{p}^P. \tag{9}$$

As indicated before, superscripts $I$ denote inviscid flow quantities, where $u_i^I = \nabla_i \phi^I$ satisfies Euler equations, with $\phi^I$ the velocity potential of the inviscid flow field, and superscripts $P$ represent perturbation flow quantities that are driven by the inviscid flow fields. Applying this decomposition to NS Equations (1) and (2), assuming a LES, and using Euler equations to eliminate part of the inviscid fields, the perturbation NS-LES equations read e.g., [4],

$$\frac{\partial u_i^P}{\partial x_i} = 0$$

$$\frac{\partial u_i^P}{\partial t} + u_j^P \frac{\partial u_i^P}{\partial x_j} = -\frac{1}{\rho}\frac{\partial \tilde{p}^P}{\partial x_i} + (\nu + \nu_T)\frac{\partial^2 u_i^P}{\partial x_j \partial x_j} - \left(\frac{\partial u_i^I}{\partial x_j}u_j^P + u_j^I\frac{\partial u_i^P}{\partial x_j}\right) + 2\frac{\partial \nu_T}{\partial x_j}S_{ij}, \quad (10)$$

where $\nu$ and $\nu_T$ are kinematic molecular and turbulent viscosity, respectively, with the latter being expressed through the Smagorinsky method as,

$$\nu_T = (C_S\Delta)^2 |\mathbf{S}|, \quad \text{with} \quad S_{ij} = S_{ij}^P + S_{ij}^I = \frac{1}{2}\left(\frac{\partial u_i^P}{\partial x_j} + \frac{\partial u_j^P}{\partial x_i} + \frac{\partial u_i^I}{\partial x_j} + \frac{\partial u_j^I}{\partial x_i}\right), \quad (11)$$

the rate of strain tensor, expressed as the sum of its perturbation $S_{ij}^P$ and inviscid $S_{ij}^I$ components, both found as a function of the corresponding velocity components. $C_S$ is the Smagorinsky coefficient, $\Delta = (\Delta x \Delta y \Delta z)^{1/3}$ the grid filter length scale, and $|\mathbf{S}| = \sqrt{S_{ij}S_{ij}}$.

Equations for the pLBM, are first derived assuming a SRT method corresponding to Equation (10), by decomposing the DFs into their inviscid and perturbation components, $f_\alpha = f_\alpha^I + f_\alpha^P$. Introducing these into in Equation (4) and subtracting the LBM equation for the inviscid flow without body forces, using the dynamic pressure, we find,

$$f_\alpha^P(t + \Delta t, \mathbf{x} + \mathbf{e_{ff}}\Delta t)) - f_\alpha^P(t, \mathbf{x})) = -\frac{\Delta t}{t_0}\{f_\alpha^P(t, \mathbf{x}) - f_\alpha^{eq}(\rho^I + \rho^P, \mathbf{u}^I + \mathbf{u}^P) + f_\alpha^{eq,I}(\rho^I, \mathbf{u}^I)\}, \quad (12)$$

where the $f_\alpha^{eq,I}(\rho^I, \mathbf{u}^I)$ satisfy Euler Equations [26]. The perturbation equilibrium DFs are then found as, $f_\alpha^{eq,P}(\rho^P, \mathbf{u}^P, \mathbf{u}^I) = f_\alpha^{eq}(\rho^I + \rho^P, \mathbf{u}^I + \mathbf{u}^P) - f_\alpha^{eq,I}(\rho^I, \mathbf{u}^I)$, i.e.,

$$f_\alpha^{eq,P} = w_\alpha\left(\rho^P + \rho_o\left(3\frac{\mathbf{u}^P \cdot \mathbf{e}_\alpha}{c^2} + \frac{9}{2}\frac{(\mathbf{e}_\alpha \cdot \mathbf{u}^P)^2 + 2(\mathbf{e}_\alpha \cdot \mathbf{u}^P)(\mathbf{e}_\alpha \cdot \mathbf{u}^I)}{c^4} - \frac{3}{2}\frac{(\mathbf{u}^P)^2 + 2\mathbf{u}^P \cdot \mathbf{u}^I}{c^2}\right)\right), \quad (13)$$

which satisfy,

$$\sum_{\alpha=0}^{n-1} f_\alpha^{eq,P} = \rho^P, \quad \sum_{\alpha=0}^{n-1} e_{\alpha i}f_\alpha^{eq,P} = \rho_o u_i^P, \quad \sum_{\alpha=0}^{n-1} e_{\alpha i}e_{\alpha j}f_\alpha^{eq,P} = p^P\delta_{ij} + \rho_o u_i^I u_j^P + \rho_o u_i^P u_j^I + \rho_o u_i^P u_j^P. \quad (14)$$

These results are then extended to the MRT using the collision operator defined in Equation (7), which yields the perturbation equilibrium moments,

$$m_1^{eq,P} = e^{eq} = \rho_0((u_x^P)^2 + (u_y^P)^2 + (u_z^P)^2 + 2u_x^P u_x^I + 2u_y^P u_y^I + 2u_z^P u_z^I)$$
$$m_9^{eq,P} = 3p_{xx}^{eq} = \rho_0(2(u_x^P)^2 - (u_y^P)^2 - (u_z^P)^2 + 4u_x^P u_x^I - 2u_y^P u_y^I - 2u_z^P u_z^I)$$
$$m_{11}^{eq,P} = p_{zz}^{eq} = \rho_0((u_y^P)^2 - (u_z^P)^2 + 2u_y^P u_y^I - 2u_z^P u_z^I)$$
$$m_{13}^{eq,P} = p_{xy}^{eq} = \rho_0(u_x^P u_y^P + u_x^P u_y^I + u_y^P u_x^I)$$
$$m_{14}^{eq,P} = p_{yz}^{eq} = \rho_0(u_y^P u_z^P + u_y^P u_z^I + u_z^P u_y^I)$$
$$m_{15}^{eq,P} = p_{xz}^{eq} = \rho_0(u_x^P u_z^P + u_x^P u_z^I + u_z^P u_x^I), \quad (15)$$

with unlisted moments being unchanged from the standard MRT formulation.

Applying a Chapman-Enskog expansion to Equation (13) would show that the pLBM solution converges to Equation (10), without the contribution of the eddy viscosity (see Appendix A). Note the presence in Equations (13) and (14) of nonlinear interaction terms between the *I* and *P* fields, which express the forcing from the inviscid flow, which is independently computed, onto the perturbation flow solved in the pLBM; note, also, there is no need for computing derivatives of the velocity.

Next, we show how the eddy viscosity terms of Equation (12) are recovered using the pLBM with a LES.

*2.4. Les Turbulence Modeling with the Perturbation LBM*

Krafczyk et al. [10] showed that the 2nd-order moments of the DFs can be expressed as,

$$P_{ij} = \sum_{\alpha=0}^{n-1} e_{\alpha i} e_{\alpha j} f_\alpha = c_s^2 \rho \delta_{ij} + \rho u_i u_j - \frac{2c_s^2 \rho}{s_{xx}} S_{ij}, \tag{16}$$

where $s_{xx}$ is a relaxation frequency for these moments. Since the 1st and 2nd terms in Equation (16)'s RHS are functions of flow quantities that can be obtained through zeroth-order moments of the DFs, the resolved rate of strain tensor can be expressed as,

$$S_{ij} = \frac{s_{xx}}{2c_s^2 \rho} \{c_s^2 \rho \, \delta_{ij} + \rho u_i u_j - P_{ij}\} = \frac{s_{xx}}{2c_s^2 \rho} Q_{ij}. \tag{17}$$

Krafczyk et al. [10] further assumed that the $Q_{ij}$'s are functions of the non-equilibrium part of the DFs, $f_\alpha^{neq} = f_\alpha - f_\alpha^{eq}$ and provided their expressions as a function of the 2nd-order MRT moments $3p_{xx}$, $p_{zz}$, $p_{xy}$, $p_{yz}$, and $p_{xz}$ (i.e., $m_{9,11,13,14,15}$). This will be detailed in the next section.

Similar to LES Equation (11), they calculated the turbulent viscosity as,

$$\nu_T = (C_S \Delta)^2 |\mathbf{S}| = \frac{s_{xx}}{2c_s^2 \rho} (C_s \Delta)^2 |\mathbf{Q}|, \qquad \text{with} \quad |\mathbf{Q}| = \sqrt{Q_{ij} Q_{ij}}. \tag{18}$$

To apply the LES with a LBM, one replaces $\nu$ by $\nu + \nu_T$, which yields a new relaxation frequency (or relaxation time $1/s_{xx}$ for the 2nd-order moments,

$$s_{xx} = \frac{1}{t_0 + t_T} \qquad \text{with} \quad t_T = \frac{1}{2}\left(\sqrt{t_0^2 + 18(C_s \Delta)^2 |\mathbf{Q}|} - t_0\right), \tag{19}$$

where $t_0$ is the relaxation time based on the molecular viscosity defined before and $t_T$ includes effects of $\nu_T$.

When applying a similar LES approach to the pLBM, the moments $P_{ij}^P$ are those given by the last Equation (14). Hence, introducing Equation (9) into Equation (17) yields an expression for the perturbation rate of strain tensor that features nonlinear interaction terms between the $I$ and $P$ fields similar to those of $P_{ij}^P$,

$$S_{ij}^P = \frac{s_{xx}}{2c_s^2 \rho}\left(c_s^2 \rho \delta_{ij} + \rho u_i^P u_j^P + \rho u_i^I u_j^P + \rho u_i^P u_j^I - P_{ij}^P\right) = \frac{s_{xx}}{2c_s^2 \rho} Q_{ij}^P. \tag{20}$$

The rate of strain tensor for the total flow is thus given by,

$$S_{ij} = \frac{s_{xx}}{2c_s^2 \rho} Q_{ij}^P + S_{ij}^I. \tag{21}$$

Therefore the $|\mathbf{Q}|$ term to use in LES Equations (18) and (19), in combination with the MRT pLBM Equations (12)–(15), is modified as follows,

$$|\mathbf{Q}| = \sqrt{R_{ij} R_{ij}}, \qquad \text{with} \quad R_{ij} = Q_{ij}^P + \frac{2c_s^2 \rho_o}{s_{xx}} S_{ij}^I, \tag{22}$$

where the $Q_{ij}^P$ terms are computed with Equation (20).

Finally, based on Equation (A14), which was derived by applying a Chapman-Enskog expansion to the equilibrium DFs of Equation (13), and replacing $\nu$ by $\nu + \nu_T$ for the LES, we find,

$$\frac{\partial u_i^P}{\partial t} + u_j^P \frac{\partial u_i^P}{\partial x_j} = -\frac{1}{\rho}\frac{\partial \tilde{p}^P}{\partial x_i} + (\nu + \nu_T)\frac{\partial^2 u_i^P}{\partial x_j \partial x_j} - \left(\frac{\partial u_i^I}{\partial x_j} u_j^P + u_j^I \frac{\partial u_i^P}{\partial x_j}\right), \tag{23}$$

which is identical to the second perturbation NS Equation (10), except for the last term, which is function of the spatial gradient of the eddy viscosity. This term is not recovered also when using the standard LBM-LES scheme detailed before and, to the authors' knowledge, no other LBM-LES method has been proposed that can do so. An alternate way of including this term in the LBM model would be to add it as a body force term, such as $B_\alpha$ in Equation (4). This, however, was not investigated further since results of applications presented hereafter agreed well with their reference solution.

## 3. Turbulent Wall Model

### 3.1. Overview

When simulating high Reynolds number flows around ocean structures (Re $\gtrsim 10^6$), with a LBM/pLBM or another computational fluid dynamics method for that matter, one typically uses a wall model to represent the flow within the thin boundary layers (BLs), near solid boundaries, rather than resolving it by refining the grid, which would be computationally prohibitive (even using an adaptive grid refinement), due to the large range of spatial and temporal scales involved. In such a model, the flow velocity is assumed to be one-dimensional, stationary, nearly wall-parallel, and well represented by a semi-empirical profile, while the LES adequately resolves the large eddies and parameterizes the SGS turbulent dissipation within the bulk of the flow, the wall model allows for an accurate representation of the large velocity gradients and corresponding shear stresses, that occur near solid boundaries as a result of the no slip boundary condition on the walls.

In this section, we detail how the LBM/pLBM models detailed before are modified, in particular how the DFs are reconstructed, at lattice nodes adjacent to solid boundaries, to include a turbulent wall model, together with a standard LBM "bounce-back scheme" e.g., [9,26]. Our method is based on that proposed by Malaspinas and Sagaut [30], who used Musker's semi-empirical profile [35] in the BLs, which combines an experimentally validated turbulent outer region, with a logarithmic profile (for $y^* \in [\delta_i, \Delta_y]$), and a viscous sublayer, with a linear shear profile close to the wall (for $y^* < \delta_i$), "patched" in a transition region (Figure 2), i.e.,

$$
\begin{aligned}
u_{x^*}(y^+) = u_\tau \bigg( & 5.424 \operatorname{atan}\left(\frac{2.0\,y^+ - 8.15}{16.7}\right) + \\
& 0.434 \log_{10}\left(\frac{(y^+ + 10.6)^{9.6}}{(y^{+2} - 8.15\,y^+ + 86.0)^2}\right) - 3.507279 \bigg),
\end{aligned}
\tag{24}
$$

where $u_{x^*}$ denotes the mean velocity component parallel to the wall, in the local direction $x^*$, and $u_\tau$ the friction velocity, and $y^+$ the non-dimensional distance from the wall in the BL, are defined as,

$$
u_\tau = \sqrt{\tau_w/\rho} \qquad \text{and} \qquad y^+ = y^* \frac{u_\tau}{\nu},
\tag{25}
$$

with the wall shear stress formally given by, $\tau_w = \mu(\partial u_{x^*}/\partial y^*)$ at $y^* = y^+ = 0$.

In addition, within the BL region of the LBM/pLBM lattices, rather than using the LES Equation (11) or (18), the turbulent eddy viscosity is parameterized using a standard model [36],

$$
\nu_T = \left(\kappa q D\right)^2 \left|\frac{\partial u_{x^*}}{\partial y^*}\right|,
\tag{26}
$$

where $\kappa$, the von Kármán constant, is set to 0.384 based on experimental data from Nagib and Chauhan [37] and $D = (1 - e^{\frac{-y^+}{A^+}})$ is a Van Driest damping function (with $D = 0$ at $y^+ = 0$), which prevents an over-prediction of the eddy viscosity near the wall, with $A^+ = 26$, the Van Driest constant.

To more accurately represent effects of highly curved boundaries on the LBM-DF reconstruction within the BL, Malaspinas and Sagaut's wall model was modified as detailed below. Indeed, a large curvature will cause rapid changes in the distance to the wall of the

LBM nodes that are closest to the wall ($q$ and $\mathbf{x}_1$, respectively, in Figure 2), which will affect the accuracy of the DF values calculated next to the wall.

The wall model will first be validated for a turbulent channel application, using LBM discretizations where nodes $\mathbf{x}_1$ are located at various distances from the wall within the logarithmic BL region (Figure 2). Then the model will be used to simulate the flow around a NACA foil, whose geometry has some highly curved areas (e.g., near the nose).

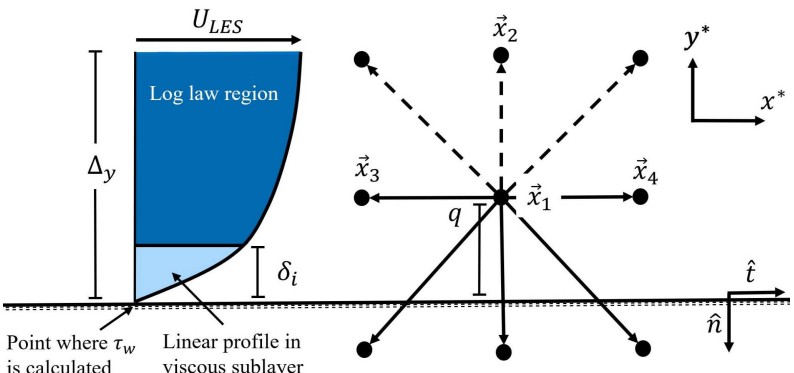

**Figure 2.** Sketch of LBM/pLBM node DF ((——) known, and (- - - -) missing) reconstruction and typical boundary layer (BL) profile e.g., [35] near a solid boundary/wall (here assumed to be 2D for simplicity). The wall parallel coordinate system is defined as $(x^*, y^*)$, based on unit normal and tangential vectors to the wall $(\hat{n}, \hat{t})$. In this sketch, $U_{LES} = \tilde{u}_2$ and $\vec{x}_i = \mathbf{x}_i$, as used in the text.

### 3.2. Combining the LBM/pLBM with the Turbulent Wall Model

The wall model velocity profile and eddy viscosity (Equations (24)–(26)) are used to reconstruct the DFs at nodes that are nearest the wall in the cross-wall direction, i.e., at locations $\mathbf{x}_1$ (distance $q$), together with velocity $\mathbf{u}_2$ computed in the LBM/pLBM solution at time $t$, at the next layer of nodes that are closest in the cross-wall direction, i.e., at locations $\mathbf{x}_2$. As is standard in the LBM "bounce back" type wall treatment, near solid boundaries, the DFs of particles moving away from the wall are assumed to be unknown after the propagation step and, hence, must be reconstructed. In Figure 2, these are the DFs for particle vectors marked by dashed lines, for which $\mathbf{e}_\alpha \cdot \hat{\mathbf{n}} < 0$, with $\hat{\mathbf{n}}$ the local outward normal unit vector to the wall.

To apply the wall model, one first solves $\tilde{u}_2 = u_{x^*}(y_2^+, \tau_w)$ for the wall shear stress $\tau_w$, using Equations (24) and (25), with $\tilde{u}_2$ the module of the total flow velocity computed with the LBM or pLBM at node $\mathbf{x}_2$ (hence not just the perturbation flow in the latter case) and tildes now indicating that flow parameters are computed using the wall model, within the BL. Then one computes $\tilde{u}_1 = u_{x^*}(y_1^+, \tau_w)$ and the eddy viscosity $\nu_{T1} = \nu_T(q, D, \partial u_{x^*}/\partial y^*)$ calculated at $\mathbf{x}_1$, using Musker's profile Equations (24) and (26). Details of numerical methods are given in the next section.

Once the flow parameters are computed in the BL, the unknown DFs are reconstructed at $\mathbf{x}_1$ and time $t$ as,

$$f_\alpha(\mathbf{x}_1, t) = f_\alpha^{eq}(\tilde{\rho}, \tilde{\mathbf{u}}_1) + f_\alpha^{neq}\left(\frac{\partial \tilde{\mathbf{u}}}{\partial \tilde{\mathbf{y}}}(\mathbf{x}_1)\right), \tag{27}$$

with $f_\alpha^{eq}$ given by Equation (5) or (13) for the LBM or pLBM implementations, respectively, $\tilde{\mathbf{u}}_1 = \tilde{u}_1\hat{\mathbf{t}}$, with $\hat{\mathbf{t}}$ the local tangential unit vector to the wall, and $\partial\tilde{\mathbf{u}}/\partial\tilde{\mathbf{y}} = -\partial\tilde{\mathbf{u}}/\partial\hat{\mathbf{n}}$. Consistent with a thin BL assumption, we set $\tilde{\rho} = \rho(\mathbf{x}_2)$.

Following Krafczyk et al. [10] and Malaspinas and Sagaut [30], and using Equation (17), we have for the LBM,

$$f_\alpha^{neq} = -\frac{w_\alpha \rho_0 D}{c_s^2 t_\nu}\{e_{\alpha i}e_{\alpha j} - c_s^2 \delta_{ij}\} S_{ij}, \tag{28}$$

where $t_\nu \sim t_0$ denotes a laminar relaxation time. For the pLBM, $f_\alpha^{neq,P}$ is calculated using $S_{ij}^P = S_{ij} - S_{ij}^I$ instead of $S_{ij}$ in Equation (28). In the BL, near the wall, the total or perturbed

rate of strain tensors are computed using flow parameters computed at $\mathbf{x}_1$ as detailed above. Similar to the eddy viscosity in Equation (26), to prevent an over-estimation of $f_\alpha^{neq}$ for small $y^+$ values near the wall (within the lower turbulent and transitional regions of the BL), the non-equilibrium DFs are multiplied by a Van Driest damping function $D$ in Equation (28).

We verified in applications that using the Van Driest damping function improves the convergence of the solution to its reference data with grid refinement, over a larger range of $y^+$ values. This is more particularly the case when there are highly curved boundaries within a regular lattice, causing rapid variations in $q$ along the boundary. We damped $f_\alpha^{neq}$ with $D$ rather than $D^2$, as for the eddy viscosity, based on a convergence test done for the turbulent channel application reported in Section 4.1. Note that, using this non-physical damping function, the shear stress is not well resolved by Equation (28), for $y^+ \to 0$. However, this stress is not used to compute forces acting on the boundary, which instead are based on the shear stress calculated using the macroscopic Musker profile Equation (24) and the Newton law of viscosity. Finally note that, we verified that if one attempted to reconstruct the unknown values of $f_\alpha^{neq}$ using some combination of $\sum_\alpha f_\alpha^{(neq)} = \sum_\alpha e_{i\alpha} f_\alpha^{(neq)} = 0$ and Equation (27) with a known velocity gradient, this would yield either an under-determined or inconsistent set of equations using a D3Q19 lattice scheme, depending on the wall orientation and number of unknowns.

### 3.3. Numerical Implementation of Wall Model in the LBM

As indicated above, to evaluate the macroscopic variables of interest at each time $t$ in the BL region of the LBM lattice (i.e., at nodes $\mathbf{x}_1$; Figure 2), Equation $\tilde{u}_2 = u_{x^*}(y_2^+, \tau_w)$ must first be solved for the wall shear stress $\tau_w$ and friction velocity $u_\tau$, using Equations (24) and (25), the flow velocity computed at current time with the LBM at $\mathbf{x}_2$, and assuming $\tilde{\rho} = \rho(\mathbf{x}_2)$. This equation is solved using Newton iterations, with $\tilde{u}_2 = |\mathbf{u}(\mathbf{x}_2, t) \cdot \hat{\mathbf{t}}|$, the local projection of the LBM-LES velocity in the (tangential) direction parallel to the boundary $\hat{\mathbf{t}}$. Once $u_\tau$ is known, the flow variables at nodes $\mathbf{x}_1$ are computed with Equation (24), i.e., $u_{x^*}(y_1^+)$ and $\partial u_{x^*}/\partial y^*(y_1^+)$ and the corresponding eddy viscosity $\nu_{T1}$ is computed with Equation (26).

Based on these flow variables, the 2nd-order DF moments (and DFs) are reconstructed at nodes $\mathbf{x}_1$ of the LBM lattice using the relaxation frequency,

$$s_{xx} = \frac{1}{t_0 + \frac{\nu_{T1}}{c_s^2}} \tag{29}$$

instead of using the LES relaxation frequency defined in Equation (19).

When applying the wall model and related methods to an arbitrary boundary geometry, a shift in reference frame is needed, such that the local $x^*$-axis points towards the local streamwise direction $\hat{\mathbf{t}}$ and locations $\mathbf{x}_1$ and $\mathbf{x}_2$ align with the outward wall-normal direction $\hat{\mathbf{n}}$ (Figure 2). Accordingly, locations $\mathbf{x}_2$ in the LBM lattice are found by first finding the direction $\alpha$ in the D3Q19 lattice yielding the largest normalized distance, $(\mathbf{e}_\alpha \cdot -\hat{\mathbf{n}})/|\mathbf{e}_\alpha|$, in the wall-normal direction, at the closest lattice nodes $\mathbf{x}_1$, with $\hat{\mathbf{n}}$ the corresponding outward unit normal vector at the wall; then $\mathbf{x}_2 = \mathbf{x}_1 + \mathbf{e}_\alpha \Delta t$ is computed. When the wall does not coincide with lattice nodes, a small (but acceptable) geometrical error will occur as no $\mathbf{e}_\alpha$ will perfectly align with the wall normal at $\mathbf{x}_1$.

To apply this method, for each selected $\mathbf{x}_1$, both $\hat{\mathbf{n}}$ and $q$ must thus be computed at/from the closest point on the boundary to $\mathbf{x}_1$. To do so, the geometry of arbitrary curved boundaries is locally approximated by a polynomial (e.g., see Section 4.2) and the minimum distance to $\mathbf{x}_1$ is found by Newton iterations, here using a maximum error of $\Delta x/12$ to limit the number of iterations. Because both the considered objects and the LBM nodes are stationary, these computations of $\hat{\mathbf{n}}$ and $q$ are done at the start of simulations ($t = 0$) for all $\mathbf{x}_1$ locations. The tangential direction $\hat{\mathbf{t}}$, however, which is aligned with the streamwise

flow, is time dependent. Following Malaspinas and Sagaut [30], it is found at each time $t$ based on the LBM-LES flow velocity computed at $\mathbf{x}_2$ as,

$$\hat{\mathbf{t}} = \frac{\mathbf{u}_2 - (\mathbf{u}_2 \cdot \hat{\mathbf{n}})\hat{\mathbf{n}}}{|\mathbf{u}_2 - (\mathbf{u}_2 \cdot \hat{\mathbf{n}})\hat{\mathbf{n}}|}. \tag{30}$$

Finally, note that for highly curved boundaries, there are situations where, for a given $\mathbf{x}_1$, the associated $\mathbf{x}_2$ also requires a separate wall model evaluation (i.e., $\mathbf{x}_2$ also has lattice links that cross the solid boundary). Special consideration of these cases is needed to avoid a race condition during parallel implementation of the model.

### 3.4. Modified Wall Model Implementation for the pLBM

To apply the wall model with the pLBM, one needs to consider the total flow, $\mathbf{u} = \mathbf{u}^I + \mathbf{u}^P$ (where $\mathbf{u}^I$ is known at time $t$ from the inviscid solution in the hybrid model) when reconstructing the BL solution, hence, $\tilde{u}_2 = |(\mathbf{u}^I + \mathbf{u}^P) \cdot \hat{\mathbf{t}}|$ at $\mathbf{x}_2$, when solving for $\tau_w$ using Musker's profile. The inviscid flow components are removed once the total solution is found. The equilibrium DFs in Equation (27) are now those given by Equation (13) and Equation (30) becomes,

$$\hat{\mathbf{t}} = \frac{(\mathbf{u}_2^I + \mathbf{u}_2^P) - ((\mathbf{u}_2^I + \mathbf{u}_2^P) \cdot \hat{\mathbf{n}})\hat{\mathbf{n}}}{|(\mathbf{u}_2^I + \mathbf{u}_2^P) - ((\mathbf{u}_2^I + \mathbf{u}_2^P) \cdot \hat{\mathbf{n}})\hat{\mathbf{n}}|}. \tag{31}$$

## 4. Applications
### 4.1. Simulation of a Turbulent Channel Flow

We assess the accuracy of the pLBM, with a LES and the turbulent wall model detailed before, for simulating turbulent flows in a horizontally bi-periodic channel in $(x, z)$, bounded by two flat plates separated by a distance $H$ in the vertical $y$ direction. Results for various Reynolds numbers and discretizations are compared with those of the DNS simulations of Hoyas and Jiménez [38], the semi-empirical profile of Musker [35], and the measurements of Dean [39]. Although this benchmark was considered before, to validate LBM-LES [30] and pLBM-LES [18,25] simulations with a wall model, here, we present a more complete investigation that considers a wider range of grid resolution, analyzes some turbulent properties of the flow, and demonstrates convergence of the friction coefficient and force applied to the plates.

Simulations are performed in a domain of $x$-length $L = 2\pi H$, $y$-height $H$, and $z$-width $W = \pi H$, with rigid boundaries specified at $y = 0$ and $y = H$, over which the turbulent wall model is applied. Periodic boundary conditions are specified in the 2 horizontal directions at $x = 0$ and $L$ (streamwise) and $z = 0$ and $W$ (cross-stream). The flow is forced through the channel using a body force term, $B_\alpha$, in the LBM Equation (4) and adding it to Equation (12) for the pLBM simulations, based on an acceleration $\mathbf{G}_\alpha = F\mathbf{e}_x$ with $F$ computed from a control volume approach as e.g., [30,40],

$$F(t) = \frac{2}{H}\{u_\tau(t)^2 + u_m(u_m - \overline{u}(t))\} \tag{32}$$

in which $\overline{u}$ and $u_\tau$ denote the instantaneous space-averaged $x$-component of the velocity and friction velocity, respectively, for the flow past the plates, with the latter given by Equation (25), based on the velocity profile in the wall model (Equation (24)), and $u_m$ is the target bulk (mean) flow velocity (i.e., averaged over $y$) obtained from [39]'s correlation between the friction Reynolds number of the flow computed as, $\mathrm{Re}_\tau = (H/2)u_\tau/\nu$ and the corresponding bulk Reynolds number $\mathrm{Re}_m = Hu_m/\nu$, i.e., $\mathrm{Re}_m = 14.641\,\mathrm{Re}_\tau^{8/7}$.

In the pLBM, a steady uniform inviscid velocity is specified in direction $x$ as, $\mathbf{u}^I = (U, 0, 0)$, computed with the wall model Equation (24) at $y = H/2$, i.e., $U = u_{x*}(y_m^+)$. The Smagorinsky coefficient used in the LES is $C_S = 0.16$ in all simulations, which is in the

middle of the range of recommended values. Each simulation is run until a fully turbulent flow, with quasi-steady mean velocity and pressure, is achieved.

Simulations are performed for three friction Reynolds number values, the first two being those also used by Malaspinas and Sagaut [30], $\text{Re}_\tau = 950, 2000$, and the third value being much larger, $\text{Re}_\tau = 20{,}000$; using the experimental correlation from Dean [39], these correspond to bulk Reynolds numbers, $\text{Re}_m = 3.7 \times 10^4, 8.7 \times 10^4$, and $1.21 \times 10^6$. Each case is simulated using 4 different LBM discretizations, $\Delta x = \Delta y = \Delta z = H/(2N)$, with $N = 10, 20, 30$, and 40. The full channel width is thus discretized with $2N$ LBM nodes in the $y$ direction.

In each simulation, the friction force $F_f$ applied to the plates in direction $x$ is computed as detailed below, with the corresponding friction coefficient defined as, $C_f = F_f/(\frac{1}{2}\rho u_m^2 A)$, with $A = 2WL$ the wetted area of the plates. The total force applied to the plates is formally defined as,

$$\mathbf{F} = \iint\limits_A \{p\hat{\mathbf{n}} + \tau_w \hat{\mathbf{t}}\}\, \mathrm{d}A. \tag{33}$$

In the present application, due to symmetry with respect to $y = H/2$, the mean normal component of the total force is zero and, in any case, its instantaneous value does not contribute to the friction force in the $x$ direction. In view of this, considering the $x$ direction and discretizing the above equation over the LBM lattice yields,

$$F_f = \sum_{i=1}^{Q} \tau_w^i \, (\Delta x)^2, \tag{34}$$

where $Q$ denotes the total number of plate boundary nodes in the lattice. In the standard LBM, $\tau_w^i$ is usually computed using the non-equilibrium component of the second-order moments defined in Equation (16). However, this approach may not be sufficiently accurate since the stress vectors acting on the boundary have to be extrapolated to it from the nearest lattice nodes [41]. Here, instead, the wall shear stress is more accurately computed using the values obtained using the wall model for each pair of nodes $(\mathbf{x}_1, \mathbf{x}_2)$ (Figure 2).

Figure 3a–c show the $x$-component of the mean velocity, averaged in the $x$ direction, $u^+ = \tilde{u}/u_\tau$, computed with the pLBM-LES model as a function of the non-dimensional distance $y^+$ across the channel, compared to Musker's profile. Results of the 4 discretizations agree well with each other and with Musker's semi-empirical profile, for each of the 3 Reynolds number values, confirming that the wall model allows accurately simulating the near-boundary flow with the pLBM, for a wide range of discretizations and, hence, $y^+$ locations of the first off-wall node in the lattice, $\mathbf{x}_1$ (about 10 to 1000). For these simulations, Figure 3d shows the bulk friction coefficient $C_f$ computed as a function of $\text{Re}_m$ with the pLBM, based on Equation (34), compared to the mean value measured by Dean [39]; the upper and lower bounds of these measurements are marked in the figure, indicating a significant experimental variance.

Figure 4 shows statistics of the resolved turbulent velocity fluctuations plotted as a function of the scaled distance from a plate, in the form of scaled Reynolds stresses $-R_{ij}/\rho = \overline{u_i' u_j'}$, i.e., $(u_i' u_j')_{rms}^+ = \sqrt{\overline{u_i' u_j'}}/u_\tau$, computed with the pLBM-LES model, with the wall model, using $N = 40$ nodes over the channel half height, for $\text{Re}_m = 37{,}042$ and $87{,}000$. Results are compared to those of the direct NS (DNS) simulations of Hoyas and Jiménez [38]. Overall, the pLBM-LES results agree well with the DNS reference data, but all the turbulent statistics are under-predicted near the plate (mostly for $y < 0.1H$), indicating that the turbulent kinetic energy $k$ of the flow is too low in the model near the wall. This is likely due to the use of Musker's mean velocity profile (Equation (24)) in the wall model, which assumes that there are no mean pressure or velocity gradient in the $x$ direction. A more advanced non-equilibrium model that does not make these hypotheses would likely improve the results.

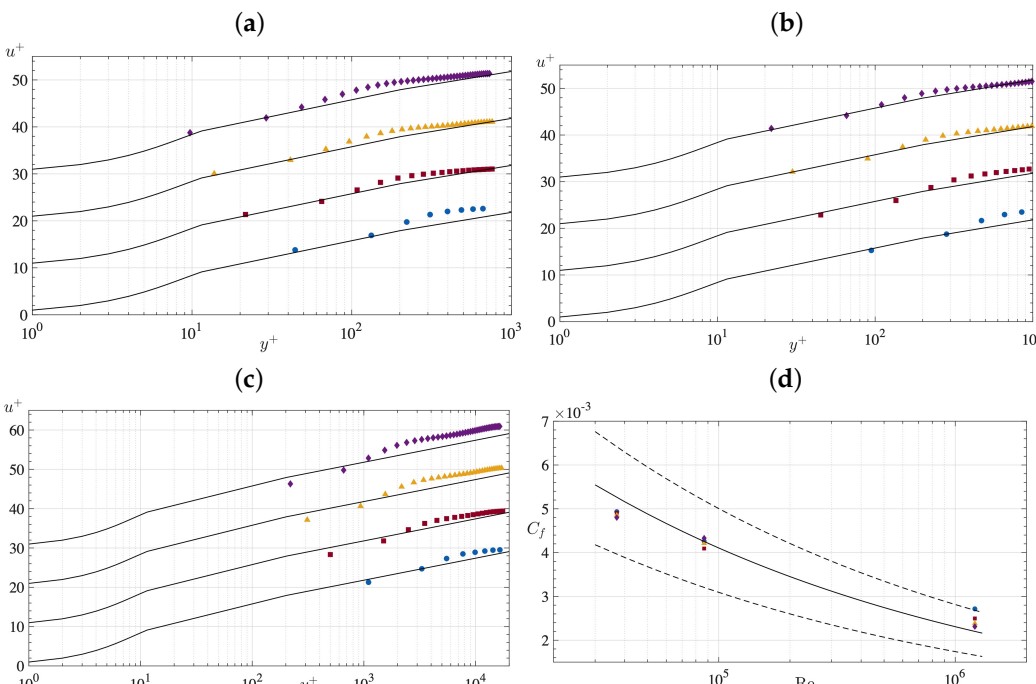

**Figure 3.** Results (symbols) of turbulent channel flow simulations. (**a–c**) Semi-log plots of mean scaled velocity $u^+ = \bar{u}/u_\tau$ as a function of the distance $y^+$ from one plate, for a half-channel, computed using the pLBM-LES with a wall model, for Re$_m$= (**a**) 37,042, (**b**) 87,000, and (**c**) $1.21 \times 10^6$, with a grid resolution $N$ = (●) 10, (■) 20, (▲) 30, and $N$ = 40 (♦). Musker's profile [35] (——) is shown for comparison. Note, for clarity, results for $N$ = 20, 30, and 40 were shifted by $\Delta u^+$ = 10, 20, and 30, respectively. (**d**) Friction coefficient $C_f$ on the plates as a function of Reynolds number, computed with the pLBM (symbols), compared to [39]'s experimental data: mean (——) and upper/lower bounds (- - - -).

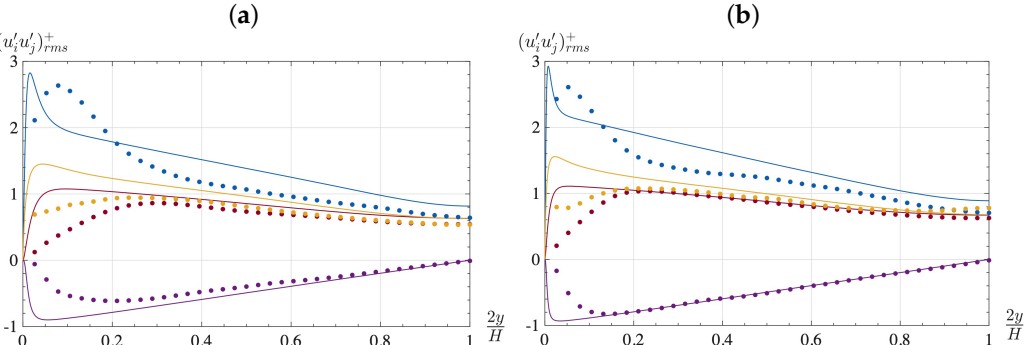

**Figure 4.** Statistics of velocity fluctuations, in turbulent channel flow simulations with: (bullets) the pLBM-LES model with the wall model and $N$ = 40, and (lines) the DNS model of [38], for Re$_m$ = (**a**) 37,042, and (**b**) 87,000. The scaled root-mean-square (*rms*) of products of velocity fluctuations, $(u'_i u'_j)^+_{rms} = \sqrt{\overline{u'_i u'_j}}/u_\tau$ is plotted as a function of the distance from a plate for: $(i = j = 1)$ (●), $(i = j = 2)$ (●), $(i = j = 3)$ (●), and $(i = 1, j = 2)$ (●).

As stated in Section 3, when Equation (28) was used without the Van Driest damping ($D = 1$) to compute the non-equilibrium DFs in the wall model [18,25,30], the solution diverged near the wall for Re$_\tau$ = 950 and 2,000, with the finest discretization and in the calculation of $C_f$. The present results, which use the complete Equation (28), show a significant improvement in the convergence of the wall model solution with the finer grid (see Figures 3c,d and 4).

Finally, for the sake of brevity, results of the LBM-LES model are not shown for this application, as they were closely identical to the results of the pLBM-LES model, both using the wall model with the same domain size, discretization, and periodic boundary conditions.

### 4.2. Simulations of Turbulent Flow around a Submerged Foil

4.2.1. Overview

We simulate the more realistic case of the turbulent flow around a three-dimensional foil, with a NACA-0012 profile in the $x$-streamwise direction, forced by a uniform inviscid free-stream velocity $\mathbf{u}^I = (U, 0, 0)$. Due to the foil highly curved boundary near its nose and large gradients of both the inviscid and perturbation fields near the foil, this is a significantly more rigorous test of the LBM/pLBM-LES models, with the wall model implementation. Simulations are performed for a high Reynolds number value, Re $= UC/\nu = 1.44 \times 10^6$ (with $C$ the foil chord) at 3 angles of attack of the foil, $\theta = 0°, 4°, 8°$.

In this application, we use versions of the models developed for nested grid simulations, and run each simulation for 4 nested grid levels (referred to as Grids 0 to 3 from coarse to fine; Table 1 and Figure 5), each with 4 different discretization $\Delta x/C$ values, in order to assess the convergence of model results with grid resolution. To ensure a smooth transition of the computed solution between nested grids, we specify a 3 lattice nodes overlap between the boundary of a domain and the boundary of a nested (finer) domain. The DFs are passed between nodes in this overlapping region of nested meshes using the method described in Filippova and Hänel [42] for the LBM and in O'Reilly et al. [26] for the pLBM. Nested grid discretization ratios used in the present application, which are listed in Table 1, vary from 4 to 2, with the reference discretization being the finest one in Grid 3, that encompasses the foil. In simulations, the Grid 3 resolution is set to $\Delta x/C = 4.0 \times 10^{-3}, 3.5 \times 10^{-3}, 3.0 \times 10^{-3}$, or $2.5 \times 10^{-3}$. A refinement ratio of 4 is often avoided as it can cause reflections of acoustic waves at the nested mesh boundaries. This ratio was required because of the large range of scales that must be resolved and due to memory limitations of the GPGPUs. Significant acoustic reflections were not observed.

**Table 1.** Grid geometry and mesh size parameters of the nested grid set-up for LBM/pLBM-LES simulations of turbulent flows around a submerged foil of chord $C$, with a NACA-0012 profile in $(x, y)$ streamwise direction, with its leading edge located at $x = y = z = 0$. LBM simulations are run in Grids 0-3, but pLBM simulations only use Grids 1–3. The foil span $S$ extends to the cross-stream boundaries ($z$-direction) of each grid, to the outermost lateral boundary.

| Grid | Min. Extent $(x, y, z)/C$ | Dimensions $(x, y, z)/C$ | Nesting Ratio |
|------|---------------------------|--------------------------|---------------|
| 0 | $(-23.7\ -30.0\ -0.3)$ | (72.0, 60.0, 0.8) | 32 |
| 1 | $(-1.85\ -1.5\ -0.1)$ | (6.0, 3.0, 0.4) | 8 |
| 2 | $(-0.45\ -0.25\ 0.0)$ | (3.0, 1.0, 0.25) | 2 |
| 3 | $(-0.1\ -0.125\ 0.025)$ | (1.7, 0.25, 0.2) | 1 |

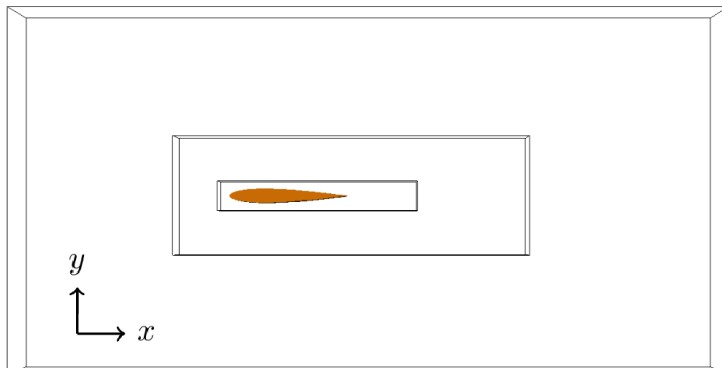

**Figure 5.** Sketch (not to scale) of nested boundaries of Grids 1–3 (black lines) with the foil marked in orange of the Nested grid set-up defined in (Table 1).

The turbulent wall model described in Section 3 is applied over the entire foil surface, so it is assumed that no laminar to turbulent transition occurs along the foil boundary during simulations, as the flow moves from the foil nose to its trailing edge. With the grid setup in Figure 5, a maximum value $y^+ = 292$ was found in the wall model, for the first lattice node off the foil boundary, when using the coarsest mesh for $\theta = 8°$, which in view of the previous application should be acceptable (see Figure 3c).

Assuming a foil chord length $C = 1$ m and that the simulations take place in air ($\nu = 1.5 \times 10^{-5} \text{m}^2 \cdot \text{s}^{-1}$), for the assumed Re value, we find the free stream velocity, $U = \nu \text{Re}/C = 21.758 \text{ m} \cdot \text{s}^{-1}$ and, assuming $c_s = 343 \text{ m} \cdot \text{s}^{-1}$ (for air at 20C), we match the simulated Mach number to its physical value as, $\text{Ma} = U/c_s = U'/c_s' = 0.063$. The associated physical time step and LBM scaling parameters can now be found as detailed in Section 2.2. The DFs were initialized to their equilibrium values for zero ambient pressure and the free stream velocity.

In the following, we compare the LBM or pLBM simulation results to wind tunnel measurements [43,44] and to results of the commonly used airfoil analysis model *Xfoil* [45]. Gregory and O'Reilly [43] measured flow parameters for NACA-0012 foils of varying roughness, at Re = $1.44 \times 10^6$, and Sheldahl and Klimas [44] performed similar experiments at Re = $1.36 \times 10^6$. In these experiments, the flow was allowed to freely transition from laminar to turbulent regimes (i.e., no trigger wires were used) and efforts were made to eliminate 3D effects such as tip vortices. *Xfoil* is a 2D BEM potential flow solver that simulates the boundary layer and trailing wake using an integral boundary layer formulation. All the *Xfoil* simulations were run to match the Mach and Reynolds numbers values of the LBM/pLBM simulations, using 200 BEM panels on the foil surface (which ensured convergence), and a turbulent boundary layer was specified along the foil boundary that developed streamwise from its leading edge.

Results of the LBM/pLBM simulations are used to compute the classical airfoil performance metrics, which are compared with the reference experimental and numerical data in the following sections. These are the pressure coefficient distribution $C_P(\mathbf{x})$ obtained from the calculated mean pressure distribution along the foil boundary $\overline{p}(\mathbf{x})$, and the lift and drag coefficients $(C_L, C_D)$ obtained from the mean hydrodynamic force applied to the foil, computed by integrating the pressure and shear stress $(p, \tau)$ along the foil surface, formally given by Equation (33), with $F_L = \overline{\mathbf{F}} \cdot \mathbf{e}_y$ and $F_D = \overline{\mathbf{F}} \cdot \mathbf{e}_x$, i.e.,

$$C_P(\mathbf{x}) = \frac{\overline{p}(\mathbf{x}) - p_a}{\frac{1}{2}\rho U^2} \quad , \quad C_D = \frac{F_D}{\frac{1}{2}\rho U^2 CS} \quad \text{and} \quad C_L = \frac{F_L}{\frac{1}{2}\rho U^2 CS}, \quad (35)$$

with $p_a$ the reference ambient pressure. In the present application, the force $\mathbf{F}$ applied to the foil is computed in the LBM and pLBM simulations using the momentum exchange method [9].

### 4.2.2. Simulations with the LBM-LES, with Turbulent Wall Model

To simulate this application with the standard LBM model, relevant boundary conditions are specified in Grid 0 (Table 1), i.e.,: (i) periodic DFs along the cross-stream/sidewall boundaries (at $z = -0.3C, 0.5C$); (ii) a free stream velocity $\mathbf{u}_\infty = (U, 0, 0)$ on the inlet/top/bottom boundaries ($x = -23.7C$, $y = \pm30C$), by way of setting the DFs to, $f_\alpha = f_\alpha^{eq}(\rho_a, \mathbf{u}_\infty)$; (iii) a zero normal gradient of the DFs, $\partial f_\alpha/\partial x = 0$ on the outlet boundary ($x = 48.3C$). Preliminary simulations, not shown here, confirmed that the domain boundaries in the $(x, y)$ directions were sufficiently far enough away from the foil, to prevent any spurious effect on the computed flow around the foil, and that the domain was wide enough in the spanwise ($z$) direction to allow for the largest eddies to develop in a 3D manner, which is particularly important to the LES.

In the simulations quasi-steady mean fields (i.e., a fully developed turbulent flow) were achieved after 3 seconds for a foil angle of attack, $\theta = 0°$ or $4°$, and 5 s for $\theta = 8°$. Simulating 1 s of the flow in the finest resolution used for Grid 3, $\Delta x/C = 2.5 \times 10^{-3}$, required 5.1 h of computations on a NVIDIA® Tesla® K80 GPU, in single precision. Further results also showed that if Grid 3's discretization was further refined, double precision calculations were required to achieve convergence in the Newton iterative scheme used to invert Equation (24) in the wall model; which led to longer computations and also to having less memory available to store the grid on the GPU. Using an explicit wall model that does not require an iterative solution could thus potentially increase our model's computational efficiency. This was proposed by [46], in their LBM-RANS simulations, who assumed a power law for the velocity profile within the entire BL. However, such BL profiles would yield less accurate results within the linear shear and transitional BL profile regions and, hence, less accurate computations of forces applied to the foil.

Figure 6 shows results of the grid convergence simulations with the LBM-LES model, for $C_L$ and $C_D$, computed as a function of the foil angle of attack using Equation (35), with the hydrodynamic forces being averaged over the last 10% of each simulation; *Xfoil* results and experimental measurements by Gregory and O'Reilly [43] and Sheldahl and Klimas [44] are shown for comparison. The figure both demonstrates a good convergence of the computed coefficients, and a good agreement of those obtained in the finest grid with the experimental measurements and with *Xfoil* results. This is more so for the lift coefficient, which essentially results from differences in the dynamic pressure distribution along the upper and lower boundaries of the foil. In the LBM model, as with any NS solver, pressure is more accurately simulated than shear stresses, which both depend on the solution in the lower BL and, in this application, are much smaller than pressure, leading to drag coefficients also smaller by an order of magnitude than the corresponding lift coefficients. This is also reflected in the experimental measurements that show, for each angle of attack, a wider range of values obtained for $C_D$ than for $C_L$, as a function of foil roughness and small differences in Re values. Both the LBM and *Xfoil* results for $C_D$ are well within this experimental uncertainty. Note that in experiments, the rough foil has larger $C_D$ values, because both skin friction is larger due to the increased roughness and the added roughness also trips a turbulent BL close to the leading edge of the foil, further increasing shear stresses and drag. In the LBM simulations, we assumed a fully turbulent BL over the entire foil. Accordingly, our calculated $C_D$ values in the finest grid resolution are in better agreement with measurements made for a smooth foil; this observation also applies to *Xfoil* results.

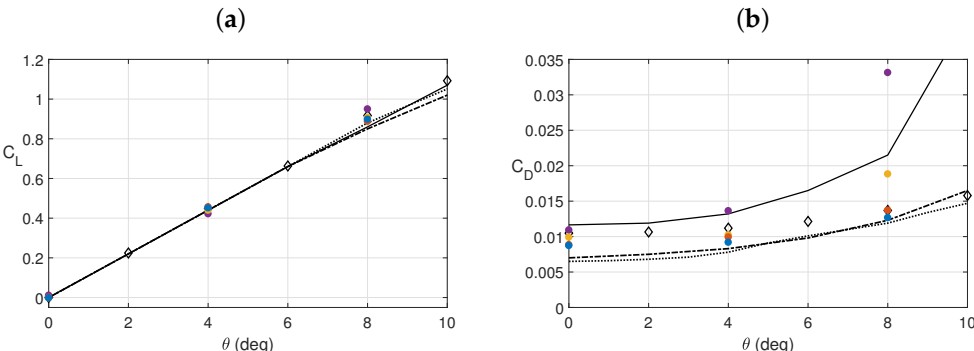

**Figure 6.** LBM-LES simulations of (**a**) lift, and (**b**) drag coefficient for a NACA-0012 foil profile, as a function of the angle of attack $\theta$, for Re = $1.44 \times 10^6$ with a Grid 3 resolution $\Delta x/C$ (Table 1): (•) $4.0 \times 10^{-3}$, (•) $3.5 \times 10^{-3}$, (•) $3.0 \times 10^{-3}$, and (•) $2.5 \times 10^{-3}$.; compared to: (◇) *Xfoil* results and measurements by [43] for Re = $1.44 \times 10^6$, for rough (—) and smooth (- - -) foils, and [44] at Re = $1.36 \times 10^6$, for a smooth foil (– – –).

Figure 7 shows the distribution of the pressure coefficient $C_P(\mathbf{x})$ along the upper and lower boundaries of the foil, at angles of attack $\theta = 0°$ and $8°$, computed with the LBM-LES model using Equation (35), in the 3 finest discretization used for Grid 3 (Table 1), compared to *Xfoil* results. The figure demonstrates a good convergence of LBM results as a function of grid resolution, with the finest grid results agreeing well with *Xfoil* results. Note that, the pressure applied to the foil surface was calculated by linearly extrapolating that computed at the lattice nodes closest to the foil boundary. As the LBM uses a Cartesian grid, however, here as in most LBM simulations of the flow around a highly curved body, small spurious oscillations of the extrapolated pressure occur near the body surface. To reduce these oscillations, a 7-point moving average filter was applied to the computed pressure before calculating $C_P$; as seen in Figure 7, however, the latter still exhibits small oscillations. Another important factor for reducing pressure oscillations, as noted by Wilhelm et al. [46], is to define the grid and foil geometry such that the smallest distance $q$ of the nearest LBM nodes to the boundary (Figure 2) is neither too small nor too large. Here we ensured that, in Grid 3, $q_{min} \simeq 0.1\Delta x$ when applying the wall model. We found that further decreasing this minimum distance increased pressure oscillations, while increasing it yielded a smoother pressure distribution, at the expense of an accurate representation of the foil geometry, and, in this application, a decrease in the drag coefficient.

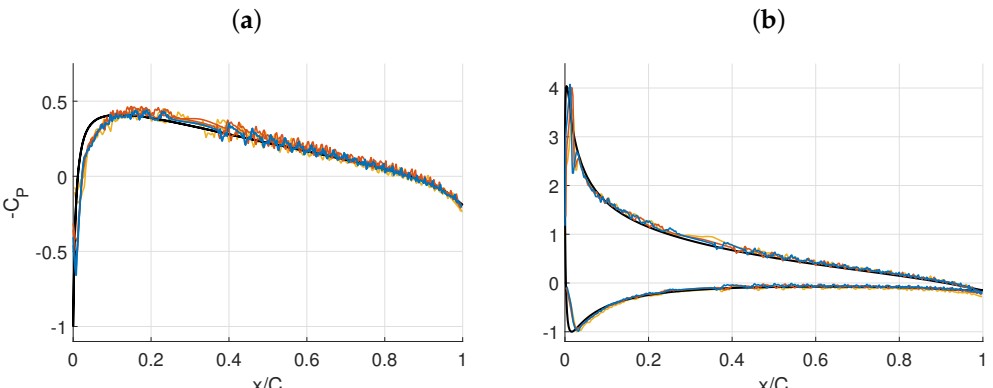

**Figure 7.** Pressure coefficients (plotted as $-C_P$) along a NACA-0012 foil upper and lower boundaries, calculated with the LBM-LES model for Re = $1.44 \times 10^6$ with a Grid 3 resolution (Table 1): $\Delta x/C = $ (—) $3.5 \times 10^{-3}$, (—), $3.0 \times 10^{-3}$, and (—) $2.5 \times 10^{-3}$, at angles of attack: (**a**) $\theta = 0°$ and (**b**) $8°$, compared to (—) *Xfoil* results.

### 4.2.3. Simulations with the pLBM-LES, with Turbulent Wall Model

To simulate this application with pLBM-LES with the wall model, the inviscid flow solution $u_i^I$ must first be computed. Here, because of the simple geometry, this is done using an analytical solution by first approximating the NACA-0012 foil by a Kármán-Trefftz foil profile, for which there exists a conformal transformation that maps the foil onto a circular cylinder, for which the analytical solution is trivial (i.e., the superposition of a uniform flow with a doublet). The inviscid solution is then used to force the perturbation flow $u_i^P$, which is solved in the pLBM. Because the far-field flow is exactly known from the inviscid solution, the pLBM model domain can be smaller than that required to apply the LBM, for a similar accuracy. Specifically, Grid 0 is not used in the pLBM simulations, and Grid 1 is slightly extended outwards, by $3C$ units, in the $x$ and $y$ directions, while this already represents a significant reduction of the domain size/number of unknowns, relative to the LBM, we believe that one could still further reduce the size of the pLBM domain, within the same result accuracy; this will be assessed in future work. For Grid 1, boundary conditions are specified as: (i) periodic DFs along the sidewall boundaries ($z = -0.1C, 0.3C$); (ii) a vanishing perturbation solution, i.e., $p^P = u_i^P = 0$ on the $x$ and $y$ extremities of the domain e.g., [26].

As indicated, the inviscid flow solution is analytically calculated by applying the Kármán-Trefftz conformal mapping function [47],

$$z = \mathcal{F}(\zeta) = \lambda a \frac{(\zeta + a)^\lambda + (\zeta - a)^\lambda}{(\zeta + a)^\lambda - (\zeta - a)^\lambda} \tag{36}$$

which transforms a circle of radius $R$ centered at $(\xi_c, \eta_c)$ in the complex plane $\zeta = \xi + i\eta$ to a foil in the complex plane $z = x + iy$, when $a = R + \xi_c$ and $\lambda = 2 - \alpha/180$, with $\alpha$ denoting the angle (in degree) of the foil trailing edge. A symmetric foil (without camber), such as a NACA-0012 foil, is obtained by setting, $\eta_c = 0$.

The Kármán-Trefftz transformation Equation (36) is first used to find the values of $a$, $R$ and $\lambda$ that yield a best fit with the NACA-0012 foil geometry. Then the corresponding complex potential, $W(\mathcal{F}(\zeta)) = \phi^I(x, y) + i \psi^I(x, y)$ of the flow around the foil (with $\psi^I$ the stream function) is computed, and the complex flow velocity, $F(z) = u(x, y) - i v(x, y)$ is finally found as,

$$F(z) = \frac{dW(\zeta(z))}{d\zeta} \left( \frac{d\mathcal{F}}{d\zeta}(\zeta(z)) \right)^{-1} \quad \text{with} \quad W(\zeta) = U \left( \zeta + \frac{R^2}{\zeta} \right). \tag{37}$$

where $\zeta(z)$ is found by inverting Equation (36). More details of this mapping method can be found in [47].

To best fit the NACA-0012 foil geometry (Figure 8a), whose trailing edge angle is $\alpha = 8.5°$, yielding $\lambda = 1.9528$, we found $(\xi_c = -0.019, 0)$ for the circle center in the $\zeta$-plane, with a radius $R = 0.273$, hence, $a = R + \xi_c = 0.254$. Using these parameters, Figure 8b (middle) shows the module of the inviscid velocity around the foil, $|F| = \sqrt{u^2 + v^2}$, calculated using Equations (36) and (37). Note that although this potential flow model provides a close fit, this Kármán-Trefftz foil has a slightly smaller chord, $0.9972C$ and thickness $0.1192C$ than the NACA-0012 foil, which has a thickness $0.12C$ for a chord $C$. Hence, there will be slight errors in the computed inviscid fields. More accurate results would be obtained by solving potential flow equations for the actual NACA-0012 foil geometry using a higher-order Boundary Element Method. This will be reported on elsewhere, see, e.g., [18,20,48].

**(a)**                                                                                                  **(b)**

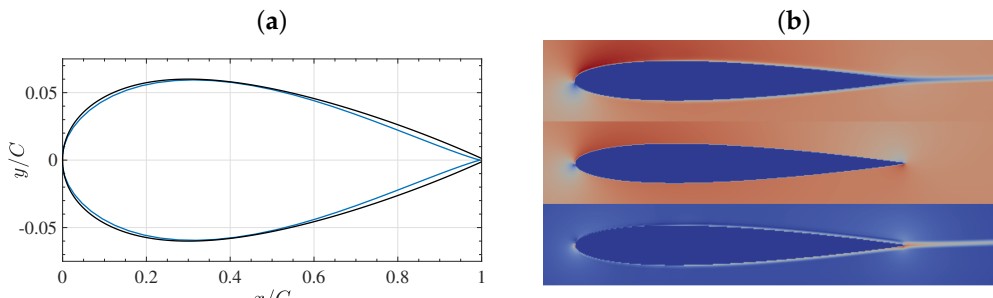

**Figure 8.** (**a**) Comparison of the Kármán-Trefftz foil profile (—), used to compute the potential (inviscid) flow component of the flow around the NACA-0012 foil profile (—-) in the pLBM. (**b**) Module of flow velocity calculated with the pLBM at steady state, for $\Delta x/C = 2.5 \times 10^{-3}$ and $\theta = 4°$: (top) total velocity, $u_i = u_i^I + u_i^P$; (middle) inviscid velocity, $u_i^I$; and (bottom) perturbation velocity, $u_i^P$.

Figure 8b shows the module of the inviscid $u_i^I$ (middle), perturbation $u_i^P$ (bottom), and total flow $u_i = u_i^P + u_i^I$ (top) components around the foil, in the finest discretization $\Delta x/C = 2.5 \times 10^{-3}$, for $\theta = 4°$. Qualitatively, the total flow appears reasonable, showing larger/lower velocities near the nose along the upper/lower boundaries of the foil. As no circulation is included in the analytical solution from Equation (37), per d'Alembert paradox, no force will be applied to the foil as a result of the inviscid flow. Hence, the pLBM solution must (and will) supply the additional perturbation flow (see Figure 8b (bottom)) that creates a circulation around the foil (equivalent to a Kutta condition at the trailing edge) and causes lift and drag forces on the foil. Note that at high angles of attack, the velocity resulting from the additional perturbation flow is in fact not small around the foil.

Figure 9 shows results of applying the pLBM model to compute the flow around the NACA-0012 foil, using 3 levels of nested grids and 4 increasingly fine discretizations, similar to the computations shown in Figure 6 for the LBM. As before, the calculated forces, and hence the lift and drag coefficients ($C_L, C_D$), are averaged over the last 10% of each simulation. Comparing the LBM and pLBM results, we see that overall the coefficients computed using the finest discretization are in good agreement with each other, and that the pLBM results agree as well with the reference data from *Xfoil* and the laboratory experiments as the LBM results, although the predicted drag coefficient may be slightly too small at the lower angles of attack. Figure 10 shows the pressure coefficients computed for the total, inviscid, and perturbation flows around the foil, with $\Delta x/C = 2.5 \times 10^{-3}$, for the cases that were simulated with the LBM (Figure 7). Similar to the LBM results, the total pressure coefficient $C_P$ is in good agreement with *Xfoil* results, although slightly worse than the LBM results near the nose on the upper boundary of the foil ($x/C < 0.13$).

**(a)**                                                                                                  **(b)**

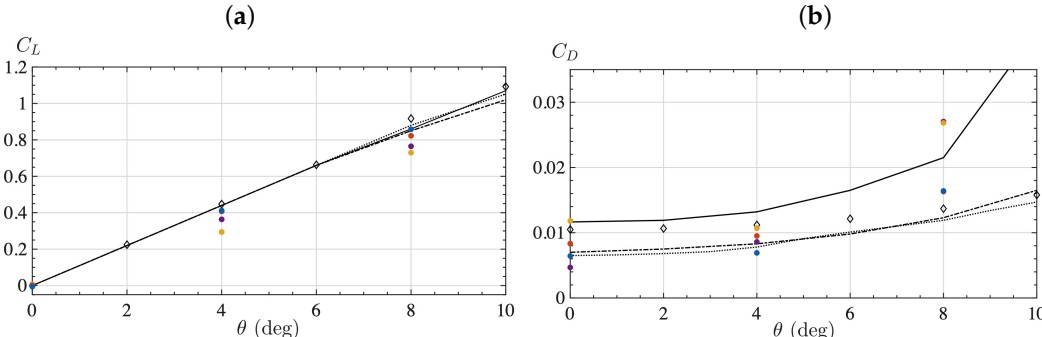

**Figure 9.** Lift (**a**) and drag (**b**) coefficient of a NACA0012 foil, as a function of its angle of attack. pLBM simulation results, calculated at Re $= 1.44 \times 10^6$, are plotted as dots, for minimum Grid 3 resolution ($\Delta x/C$) of: $4.0 \times 10^{-3}(\bullet), 3.5 \times 10^{-3}(\bullet), 3.0 \times 10^{-3}(\bullet), 2.5 \times 10^{-3}(\bullet)$. *Xfoil* simulation results are plotted as black diamonds ($\diamond$), the measurements of [43] for Re $= 1.44 \times 10^6$ for a rough foil (—), and smooth foil (- - -), and the measurements of [44] at Re $= 1.36 \times 10^6$ for a smooth foil (– - –).

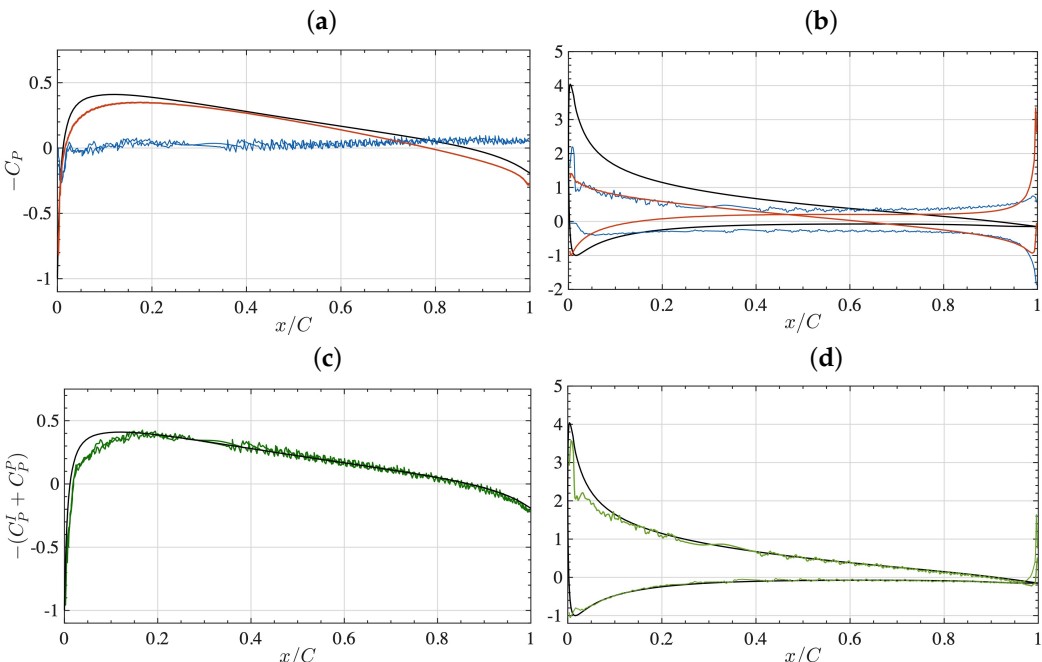

**Figure 10.** Pressure coefficients (plotted as $-C_p$) along the NACA-0012 foil upper and lower boundaries, calculated with the pLBM-LES model for Re = $1.44 \times 10^6$ with a Grid 3 resolution $\Delta x/C = 2.5 \times 10^{-3}$, at angles of attack: (**a,c**) $\theta = 0°$, and (**b,d**) $8°$, compared to *Xfoil* results (—). (**a,b**) perturbation/inviscid components, $C_P^P$ (—)/$C_P^I$ (—), and (c,d) total coefficient, $C_P = C_P^I + C_P^P$ (—).

The slight underprediction of $C_L$ and $C_P$ in the pLBM results, for the latter near the foil nose, is likely a result of using an inviscid flow solution corresponding to the slightly different Kármán-Trefftz foil, rather than the actual NACA-0012 foil geometry. As shown in Figure 8a, the former foil is thinner than the NACA foil near both its leading (nose) and trailing edges. This likely results in a slight underprediction of $u_i^I$ near the leading edge, which leads to an underprediction of the maximum of $-C_P$ in this area (Figure 10c,d), which affects the lift coefficient. Near the trailing edge of the foil, with a non-zero angle of attack, the high inviscid velocities (Figure 8b) require a larger correction from the perturbation solution, to provide circulation. This is also true for the pressure, as seen in Figure 10b,d, where there are large perturbation components near the trailing edge. Hence, a more accurate representation of the NACA foil geometry, and corresponding inviscid flow fields, would likely improve the solution in these areas and the resulting force coefficients. As indicated before, such a solution could be obtained using a BEM model.

Nevertheless, the results presented here demonstrate that the pLBM can accurately compute the flow around and forces applied to the foil, when forced by an (even slightly approximate) inviscid flow solution. The pLBM solves more complicated Equations (due to the inviscid forcing terms), however, in a smaller domain and hence with an improved computational efficiency. Using the finest resolution grids, the pLBM required ~4.8 h to compute 1 s of simulations on the same hardware as used for the LBM, which only represents a slight speed-up. However, steady-state results were achieved after only 2 s whereas it took 3 s to do so using the LBM, which means, in this application, the overall speed-up resulting from using the pLBM was 37%. The computational cost of applying the pLBM is significantly higher than that of the 2D potential flow solver Xfoil. However, the NS solution found here is general and, hence, applicable to a large class of problems with solid boundaries and complex 3D geometries; not just to foils. This is of interest when simulating marine and naval hydrodynamics systems and vehicles It is expected that much more significant computational benefits of the hybrid approach would be achieved if the pLBM was used to model free surface flows. In such cases, the far-field wave propagation would be modeled in the potential flow solution, further reducing the pLBM's computational

domain and reducing the computationally costly free surface interface calculations that would otherwise be required without a hybrid approach.

## 5. Conclusions

In this paper, a hybrid-LBM model proposed earlier and applied to DNS simulations of flows around structures at moderate Reynolds number values was extended to apply to high Reynolds number fully turbulent flow, by implementing a LES with a wall model. The hybrid solution performed in the pLBM model is based on a Helmholtz decomposition of the total flow with inviscid and perturbation flow components. In the presented applications, the inviscid/potential flow solution is computed analytically, but this could be done efficiently using a BEM model. To implement the LES in the pLBM equations, we show that perturbation DFs can be used and provide proof that the solution converges to the that of the perturbation Navier–Stokes equations.

Results show that even for fully turbulent flows in a channel or around a foil, the perturbation solution is able to provide the large correction to the potential flow required to achieve an accurate solution for the total flow. In the foil simulations, in particular, accurate lift coefficients are obtained for a range of angles of attack, without the need to supply a circulation to the inviscid solution (e.g., through a Kutta condition at the trailing edge). This indicates the possibility for the pLBM to correct an inviscid solution in future applications where 3D effects would be larger and for instance tip vortices and viscous spanwise flows would become more important. While only a small 37% computational speed-up relative to the LBM was achieved in the application of the pLBM to a NACA-0012 foil, which still is a fairly 2D flow, no attempt was made to assess whether the computational domain dimensions could be reduced without significantly affecting the pLBM results, which could further increase the computational speed-up. This will be assessed in future work, also using a more accurate BEM-based inviscid solution. Finally, higher speed-ups would be expected in other types of applications, where the bulk of the computations would not be devoted to applying the wall model to the lattice nodes located near the foil and also for truly 3D cases (e.g., underwater vehicles), for which the reduction in computational domain size would also apply to the cross-stream ($z$) direction. Further speed-up is also anticipated for applications with free surface effects as the perturbation solver will no longer need to propagate waves over large distances away from a submerged body.

**Author Contributions:** Conceptualization, C.M.O., S.T.G., C.F.J. and J.C.H.; Investigation, C.M.O.; Software, C.M.O. and C.F.J.; Supervision, S.T.G. and J.M.D.; Validation, C.M.O.; Writing—original draft, C.M.O. and S.T.G.; Writing—review and editing, C.M.O., S.T.G., C.F.J., J.M.D. and J.C.H. All authors have read and agreed to the published version of the manuscript.

**Funding:** C.M.O., S.T.G. and J.M.D. gratefully acknowledge support for this work from grants N-00014-13-10687 and N-00014-16-12970 of the Office of Naval Research (PM Kelly Cooper), as well as the support from the NSF-XSEDE grant ENG-17-0010 regarding advance GPU computational resources.

**Institutional Review Board Statement:** Not applicable.

**Informed Consent Statement:** Not applicable.

**Data Availability Statement:** Not applicable.

**Conflicts of Interest:** The authors declare no conflict of interest.

## Appendix A. Convergence of pLBM Simulations towards Results of the Perturbation NS Equations

*Appendix A.1. CE Expansion*

In the LBM literature, the Chapman-Enskog (CE) perturbation expansion has been used as a standard tool to demonstrate the macroscopic behavior of LBM model formulations e.g., [9,26,49,50]. In CE expansions, the expansion parameter $\epsilon$ is usually proportional

to the ratio of the lattice grid size $\Delta x$ to a characteristic macroscopic length (e.g., $\ell$). In the following, the CE analysis is applied to the *perturbation equilibrium particle density functions* (pEPDFs), defined in Equation (13) for the pLBM, to shows that these indeed yields a solution that converges towards that of the perturbation NS Equation (10) (except for their last term, as noted before).

Let us first consider the following quantities and scales. To simplify notations, time derivatives $\partial/\partial t$ are denoted by $\partial_t$ and spatial derivatives $\partial/\partial x_i$ by $\nabla_i$. Assuming that, $\epsilon = \Delta x/\ell \ll 1$, the particle density functions (DFs) are expanded as,

$$f_\alpha = f_\alpha^{(0)} + \epsilon f_\alpha^{(1)} + \epsilon^2 f_\alpha^{(2)} + \mathcal{O}(\epsilon^3)$$

$$\partial_t = \epsilon \partial_{t_1} + \epsilon^2 \partial_{t_2} + \mathcal{O}(\epsilon^3) \quad \text{and} \quad \nabla_i = \epsilon \nabla_i \tag{A1}$$

With these definitions, the Taylor series expansion in time and space of the LHS of Equation (4) reads,

$$
\begin{aligned}
f_\alpha(t + \Delta t, x_i + e_{\alpha i}\Delta t) &= f_\alpha(t, x_i) + \Delta t\, \epsilon (\partial_{t_1} + e_{\alpha i}\nabla_i) f_\alpha(t, x_i) + \\
&\quad \frac{\Delta t^2}{2} \epsilon^2 (\partial_{t_1} + e_{\alpha i}\nabla_i)(\partial_{t_1} + e_{\alpha j}\nabla_j) f_\alpha(t, x_i) + \mathcal{O}(\epsilon^3),
\end{aligned} \tag{A2}
$$

Introducing Equations (A1) and (A2) in Equation (4) and collecting terms of different orders yields, to the first-order (zeroth-order in $\epsilon$) [9,50],

$$\mathcal{O}(1): \qquad 0 = -\frac{\Delta t}{\tau}(f_\alpha^{(0)} - f_\alpha^{eq}), \tag{A3}$$

hence $f_\alpha^0 = f_\alpha^{eq}$.

Defining the operator $D_\alpha = \partial_{t_1} + e_{\alpha i}\nabla_i$, the particle DF components of $\mathcal{O}(\epsilon)$ and $\mathcal{O}(\epsilon^2)$ are then defined as the non-equilibrium components of the particle DFs, $f_\alpha^{neq}$, i.e.,

$$\mathcal{O}(\epsilon): \qquad D_\alpha f_\alpha^{(0)} = -\frac{1}{\tau} f_\alpha^{(1)} \tag{A4}$$

$$\mathcal{O}(\epsilon^2): \qquad \partial_{t_2} f_\alpha^{(0)} + \frac{\Delta t}{2}\partial_{t1} D_\alpha f_\alpha^{(0)} + \frac{\Delta t}{2}e_{\alpha i}\nabla_i D_\alpha f_\alpha^{(0)} + D_\alpha f_\alpha^{(1)} = -\frac{1}{\tau} f_\alpha^{(2)}. \tag{A5}$$

Substituting Equation (A4) into Equation (A5) yields,

$$\mathcal{O}(\epsilon^2): \qquad \partial_{t_2} f_\alpha^{(0)} + \left(1 - \frac{\Delta t}{2\tau}\right) D_\alpha f_\alpha^{(1)} = -\frac{1}{\tau} f_\alpha^{(2)}. \tag{A6}$$

*Appendix A.2. Particle DF Moments*

Computing the zeroth-order moment of Equation (A4) and using the pEPDFs from Equation (13) recovers the conservation of mass equation for the perturbation NS equations,

$$\sum_{\alpha=1}^{n} \partial_{t_0} f_\alpha^{(0,P)} + \sum_{\alpha=1}^{n} e_{\alpha i}\partial_i f_\alpha^{(0,P)} = -\frac{1}{\tau}\sum_{\alpha=1}^{n} f_\alpha^{(1,P)}$$

$$\partial_t \rho^P + \rho_o \nabla_i u_i^P = 0, \tag{A7}$$

while the inviscid mass conservation equation is recovered when the inviscid form of the EPDFs are used [26],

$$\rho_o \nabla_i u_i^I = 0. \tag{A8}$$

Taking the first-order moment of Equation (A4) and using the pEPDFs from Equation (13) recovers the leading order terms of the perturbation NS equations,

$$\sum_{\alpha=1}^{n} e_{\alpha i} \partial_{t_0} f_{\alpha}^{(0,P)} + \sum_{\alpha=1}^{n} e_{\alpha i} e_{\alpha i} \nabla_i f_{\alpha}^{(0,P)} = -\frac{1}{\tau} \sum_{\alpha=1}^{n} e_{\alpha i} f_{\alpha}^{(1,P)}$$

$$\partial_t \rho_o u_i^P + \nabla_i (p^P + \rho_o u_i^I u_j^P + \rho_o u_i^P u_j^I + \rho_o u_i^P u_j^P) = 0, \quad \text{(A9)}$$

and the inviscid momentum conservation Equations (Euler equations) are recovered when the inviscid form of the EPDFs of Equation (5) are used [26],

$$\partial_t \rho_o u_i^I + \nabla_i (p^I + \rho_o u_i^I u_j^I) = 0. \quad \text{(A10)}$$

The latter confirms that Euler equations are exactly represented in the LBM when using the inviscid form of the EPDFs in Equation (5), $f_{\alpha}^{eq,I}$. This is unlike NS or perturbation NS equations, in which non-equilibrium components of the EPDF's must be included to represent viscous effects. Therefore, in the hybrid modeling context, this implies that an inviscid potential flow field satisfying Euler equations can be mapped to the LBM variables using $f_{\alpha}^{eq,I}$. Finally, this confirms that the decomposition method used to derive Equations (13) does not need to consider $f_{\alpha}^{neq,I}$ terms or their moments, since these are zero by definition.

Based on these conclusions, one may infer that the numerical kinematic viscosity of the pLBM can be selected as identical to that obtained from the CE of the standard LBM. This is confirmed by taking the first-order moment of Equation (A6), and then applying Equation (13),

$$\sum_{\alpha=1}^{n} e_{\alpha} \partial_{t2} f_{\alpha}^{(0,P)} + \sum_{\alpha=1}^{n} e_{\alpha} \left(1 - \frac{\Delta t}{2\tau}\right)(\partial_{t_0} + e_{\alpha i} \nabla_i) f_{\alpha}^{(1,P)} = -\frac{1}{\tau} \sum_{\alpha=1}^{n} e_{\alpha} e_{\alpha} f_{\alpha}^{(2,P)} \quad \text{(A11)}$$

where the first moment of $f_{\alpha}^{(1,P)}$ is zero in the absence of a body force, and its second moment found by considering, $\epsilon \Pi^{(1,P)} = \Pi - \Pi^{(0,P)}$, with,

$$\Pi^{(1,P)} = \sum_{\alpha=1}^{n} e_{\alpha i} e_{\alpha j} f_{\alpha}^{(1,P)} = -c_s^2 \tau (\partial_i \rho_o u_j^P + \partial_j \rho_o u_i^P), \quad \text{(A12)}$$

and giving,

$$\partial_{t2} \rho_o u_i^P - \nabla_i \left(\tau - \frac{\Delta t}{2}\right) c_s^2 (\nabla_j \rho_o u_k^P + \nabla_k \rho_o u_j^P) = 0. \quad \text{(A13)}$$

The perturbation momentum conservation equations is finally recovered to within $\mathcal{O}(\epsilon^2)$ and $\mathcal{O}(\text{Ma}^2)$, from Equations (A9) and (A11) as,

$$\partial_t \rho_o u_i^P + \rho_o \nabla_j (u_i^P u_j^P + u_i^P u_j^I + u_i^I u_j^P) = -\nabla_j p^P + \nu \nabla_j^2 u_i^P \quad \text{(A14)}$$

when the viscosity is defined as,

$$\nu = \left(\tau - \frac{\Delta t}{2}\right) c_s^2, \quad \text{(A15)}$$

which confirms that the standard LBM relaxation time is suitable for use in the pLBM.

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
