# Peer review of "Hybrid Lattice-Boltzmann-Potential Flow Simulations of Turbulent Flow around Submerged Structures"

_jmse, doi:10.3390/jmse10111651_

Round 1

Reviewer 1 Report

Journal of Marine Science and Engineering -JMSE (ISSN 2077-1312)

Manuscript ID: jmse-1968858

Comments and Suggestions for Authors

The authors investigated the hybrid simulation of turbulent flow interactions with submerged structures in this study. The authors created a 3D hybrid LBM/LES to model incompressible turbulent ocean flows. In LBM, velocity and pressure are the sums of inviscid flow and viscous perturbation. Far-to-near-field flow is inviscid and modeled with BEM (BEM). Navier-Stokes (NS) equations model, near-field perturbation flow around structures, using LBM and LES. In this work, the authors develop a modified LBM collision operator to mimic viscous perturbation flow, resulting in pLBM. The pLBM uses LES and a wall model to simulate turbulence near solid barriers. Modeling turbulent flows across a flat plate for moderate to large Reynolds numbers confirms the hybrid model; the plate friction coefficient and near-field turbulence parameters agree with measurements and direct NS simulations. The authors simulate flow past a NACA-0012 foil using LBM-LES and the hybrid pLBM-LES model with Re = 1.44106$. Tests and numerical approaches agree on the lift, drag, and foil pressure. The pLBM model achieves similar or slightly better results than the traditional LBM in a smaller computational domain and at a lower cost, showing the benefits of the innovative hybrid technique.

The findings in this study are complete, rich, and diverse. According to the authors' presentation, twenty exciting figures are presenting their findings. The reviewer is impressed with the authors' results in this investigation. Besides, please take note of the below comments:

-        Title: Please shorten the title. It should be within ten words. Articles with long titles should not be interested in citations for journals rules for additional charges of the limited pages.

-        Citations in the text should make an order. It should start from [1], [2], …. In this manuscript, you started from [24].

-        Figure 5b: please scale the drawing bigger.

-        Please format Figure 5a in a table.

-        Line 425: why did you choose to use the coarsest mesh? Will it affect the results and accuracy of computation?

-        Please pay attention to the units. m/s should be m.s-1 or m2/s should be m2.s-1 etc.

-        Section 4.2.2 and Section 4.2.3 (Simulations with the pLBM-LES, with turbulent wall model) are the same. Please revise.

Questions:

-        What kind of mesh were you generating in this study? Do the differences in the mesh types affect the numerical results? Have you taken the mesh independence test? ? Does the thickness of the first layer of the boundary layer in the problem satisfy y+=1?

-        How many are the model's total number of meshes and nodes? Why did you choose pLBM-LES, with a turbulent wall model, for this study? Have you compared its accuracy with that of other models?

-        Have the authors compared these results to those of other authors? If not, please add some comparisons with previous studies to see the improvement of this study.

-        What is the limitation of this study? What is the authors’ further research?

The reviewer hopes that his point of view could help the authors improve their work well.

I appreciate your work.

Sincerely yours,

The reviewer

Reviewer 2 Report

Comments and suggestions in the attached file.
